# Coherent vibrational dynamics reveals lattice anharmonicity in organic–inorganic halide perovskite nanocrystals

Tushar Debnath [1] ✉, Debalaya Sarker[2], He Huang[1], Zhong-Kang Han[2], Amrita Dey[1], Lakshminarayana Polavarapu [1] ✉, Sergey V. Levchenko [2] ✉ & Jochen Feldmann[1]

The halide ions of organic-inorganic hybrid perovskites can strongly influence the interaction between the central organic moiety and the inorganic metal halide octahedral units and thus their lattice vibrations. Here, we report the halide-ion-dependent vibrational coherences in formamidinium lead halide ($FAPbX_3$, X = Br, I) perovskite nanocrystals (PNCs) via the combination of femtosecond pump–probe spectroscopy and density functional theory calculations. We find that the $FAPbX_3$ PNCs generate halide-dependent coherent vibronic wave packets upon above-bandgap non-resonant excitation. More importantly, we observe several higher harmonics of the fundamental modes for $FAPbI_3$ PNCs as compared to $FAPbBr_3$ PNCs. This is likely due to the weaker interaction between the central FA moiety and the inorganic cage for $FAPbI_3$ PNCs, and thus the $PbI_6^{4-}$ unit can vibrate more freely. This weakening reveals the intrinsic anharmonicity in the Pb-I framework, and thus facilitating the energy transfer into overtone and combination bands. These findings not only unveil the superior stability of Br–based PNCs over I–based PNCs but are also important for a better understanding of their electronic and polaronic properties.

[1] Chair for Photonics and Optoelectronics, Nano-Institute Munich, Physics Department, Ludwig Maximilians-Universität (LMU), Munich, Germany. [2] Center for Energy Science and Technology, Skolkovo Institute of Science and Technology, Moscow, Russia. ✉email: t.debnath@physik.uni-muenchen.de; l.polavarapu@physik.uni-muenchen.de; s.levchenko@skoltech.ru

Organic–inorganic halide perovskite nanocrystals (PNCs) are gaining increasing attention in contemporary research due to their promising optoelectronic performance[1–5]. Photoexcitation of these PNCs with an ultra-short laser pulse can produce coherent phonons along the lattice displacement coordinate, leading to lattice vibrations[6–13]. The soft nature of the PNCs, alongside their polar characteristics, can give rise to a strong electron–phonon coupling (governed by the Fröhlich interaction) with major consequences on their optoelectronic properties[10–13]. For instance, this can influence the intrinsic charge carrier mobilities, as well as cause inhomogeneous broadening of the optical transitions[10,14,15]. Such interactions also govern the formation and decay of the charge-localized polarons[16,17]. The polarons in perovskites are energetically stabilized by their dipolar interaction with the central organic moiety[18], which shields them from other ultrafast scattering processes. Previous studies have shown that the dynamics of organic moieties play a crucial role in the polaronic properties of hybrid perovskites[19–22]. Despite recent studies on the ultrafast carrier dynamics of different PNCs[23–29], the influence of electronic and vibrational coupling on the photophysical properties is currently under debate. However, to get a comprehensive insight, it is important to unravel how the type of both central organic moieties and inorganic metal halide octahedral cages individually contribute to the combined lattice dynamics. Very recently, excited–state vibrational features in a bulk film of MAPbX$_3$ (X = Br, I), as well as in two–dimensional lead halide perovskites have been explored by ultrafast pump–probe strategy[18,30–32]. However, lattice vibrations in colloidal PNCs have not been studied to date. It would be interesting to explore such phenomena in more stable formamidinium (FA)–based PNCs, where the size and dynamics of the organic moiety can play crucial roles. The dependence of the rotational energy barrier of the central FA moiety and lattice vibrations of the inorganic cage on the nature of halide substitutions can open new fundamental insights into the structural dynamics in such PNCs.

Herein, we have utilized optical pump–probe spectroscopy to investigate the coherent lattice vibrational dynamics for hybrid halide PNCs (FAPbX$_3$ PNCs, FA: formamidinium and X: halogen) in the low frequency (<200 cm$^{-1}$) regime. We observe pronounced energy and amplitude modulations of the differential absorption ($\Delta A$) in the time domain on an above–bandgap photoexcitation of FAPbX$_3$ PNCs by an ultrashort pump pulse. This is due to the formation of vibrational wave packets. Here the identities of the different vibrational modes during participation of the ultrafast lattice vibrational dynamics can be revealed by the assignment of the resonance occurring in the corresponding Fourier transform (FT) spectra. Ground–state first-principles calculations in the framework of the density functional theory (DFT) have been carried out to corroborate the role of central FA moiety in the experimentally observed changes in vibrational coherences upon halide substitution. The insights not only unravel the underlying reason for the halide-dependent stability of these materials but also shed light on their charge-carrier mobility and polaronic properties.

## Results

### Characterization of the FAPbBr$_3$ PNCs. 

The prototype material under the investigation, colloidal FAPbBr$_3$ PNCs, has been synthesized using the high–temperature hot–injection method previously reported in the literature[33] and is described in the "Methods" section. Fig. 1a (and Supplementary Fig. 1) shows the corresponding transmission electron microscopy (TEM) image of the synthesized pristine bromide material. As evident from the TEM image, the PNCs are monodispersed particles having cubic

morphology with ~12 nm edge length. Fig. 1b shows the optical absorption–photoluminescence (PL) spectra of the colloidal FAPbBr$_3$ PNCs. The optical spectrum is characterized by the appearance of absorption due to the exciton transition at 506 nm while the maximum in PL occurs at 517 nm.

### Vibrational wave-packet dynamics. 

Fig. 2a illustrates a scheme of the pump–probe experiment on PNCs where the ultrashort pump pulse first interacts with the PNCs, and a spatially and temporally overlapped probe pulse monitors the change in transmission with a time delay. The chopper placed in the pump path blocks every alternative pulse to read out the changes in the probe transmission for each pump–probe cycle. The pump–probe measurements were carried out by the excitation of colloidal FAPbBr$_3$ PNC solution with a non–resonant (above-bandgap) 400 nm laser, as indicated by a blue arrow in Fig. 2b. A 2D color map of the differential absorption ($\Delta A$) spectra acquired as a function of time delay is shown in Fig. 2c (also Supplementary Fig. 2). A strong negative differential absorption can be seen with a peak centered at 507 nm due to phase–space–filling (known as the ground state bleaching) of the electrons and holes. The negative differential absorption signal observed during the first picosecond gains in intensity due to the hot carrier thermalization[34]. Once the carrier thermalization saturates (~1 ps), the bleaching intensity starts reducing in the next few picoseconds, indicating the depopulation of the excitonic states (Fig. 2d). On top of the carrier dynamics, the negative differential absorption signal (bleaching) shows a periodic oscillation in the time domain. As depicted in Fig. 2d, the oscillations in the pump–probe signals acquired at the bleach maximum (510 nm) can be clearly visualized in the residual time trace obtained after the subtraction of the exponential decay component. These oscillations are generally attributed to coherent phonons arising from the lattice displacement upon excitation with a pump laser having pulse duration shorter than the period of vibrational modes[35–38]. As the spectral width of the laser pulse can cover several vibrational states of the crystal lattice simultaneously, these states are excited coherently at $t = 0$ (Fig. 2e) and the respective oscillation components are

$$e^{i(n.\triangle\omega).T} \tag{1}$$

where $n = 1, 2, \ldots$ are the coherently excited vibrational quantum states, and $\Delta\omega = \frac{2\pi}{T}$ where $T$ is the period of oscillation. In principle, the superposition of all the coherently excited states leads to the temporal evolution of vibrational wave packets, which causes spectral modulations of the electronic transitions. The interaction between the pump pulse and the crystal lattice results in the generation of wavepacket in the ground, as well as excited-state potential, which can be retrieved from the bleaching and excited-state absorption (or, simulated emission) signals, respectively[38]. The wavepacket then starts to oscillate centering at the equilibrium position of the ground state, as depicted in Fig. 2e, and gets detected by the interaction with a time-delayed probe pulse at the bleach position. This results in modulation of the probe transmission at the frequencies of the lattice vibrations, and appeared as the time domain oscillations in the $\Delta A$ signal of the pump–probe measurement, which is always superimposed with the transient-absorption (TA) signal. The TA signal is then subtracted to obtain the residual time trace, as depicted in Fig. 2d, and this reveals the oscillating temporal dynamics. The Fourier transformation of the residual oscillating signals results in the underlying vibrational frequencies of an inorganic lattice. The interpretation of these fingerprint vibrational frequencies of the inorganic cage can give insights into their interaction with

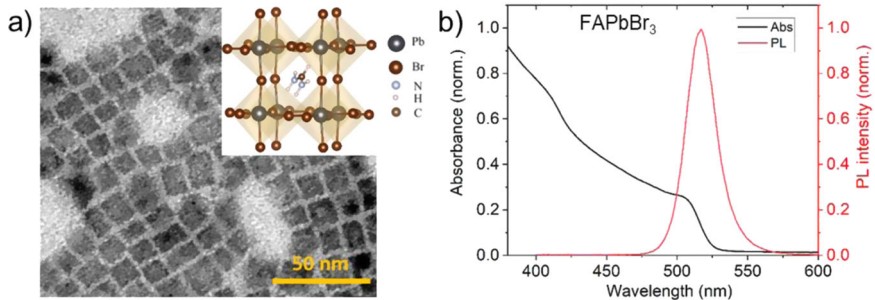

**Fig. 1 Morphological and optical properties of FAPbBr₃ PNCs. a** TEM image of the FAPbBr₃ PNCs. (Inset) Crystal structure of FAPbBr₃. **b** Steady-state optical absorption and PL spectra of the FAPbBr₃ PNCs.

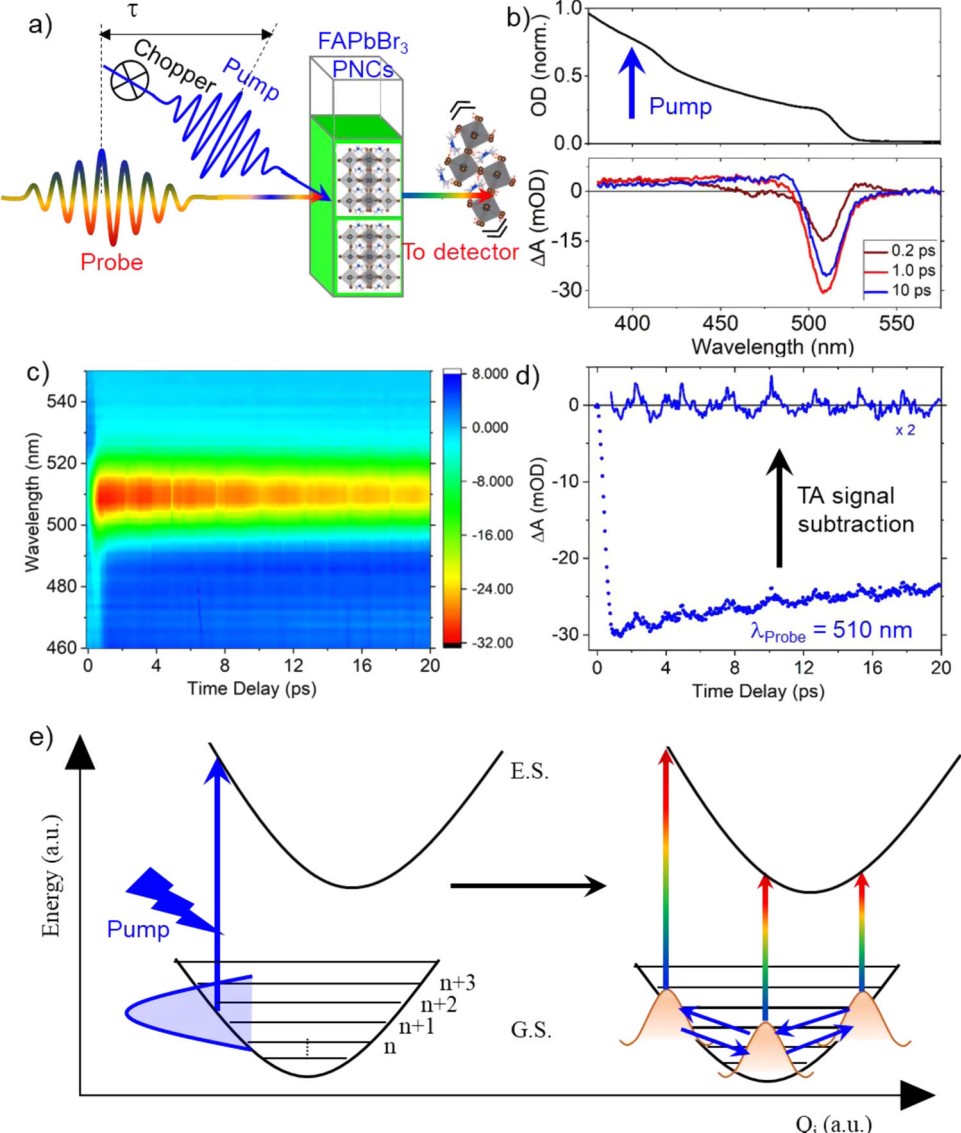

**Fig. 2 Vibrational wave-packet dynamics. a** Schematic of the ultrafast pump–probe experiment on FAPbBr₃ PNCs. **b, c** Chirp-corrected transient differential absorption ($\Delta A$) spectra of the FAPbBr₃ PNCs depicted at specific time delays of 0.2, 1, and 10 ps, and in a contour diagram, respectively. The color bar in **c** represents $\Delta A$ (mOD). The linear absorption spectrum is also shown on the top panel of **b** for comparison. The PNCs are excited with a ~100 fs laser pulse at 400 nm (excitation photon density ~3.5 × 10¹³ cm⁻²). **d** The time trace obtained at 510 nm is shown as a blue dotted line. The residual time trace, obtained after subtraction of the exponential fit, is also shown as a blue line. **e** Illustration of a vibrational wave packet motion along a particular phonon displacement coordinate of PbBr₆⁴⁻ octahedral, generated by the interaction of the pump pulse with the PNCs. The vertical and horizontal axes denote the potential energy and phonon displacement coordinate $Q_i$, respectively. The interaction of the above bandgap pump pulse with the PNCs leads to the generation of wave packets in the ground state (G.S.), as well as the excited state (E.S., not shown). Subsequently, the wave packet starts oscillating at the ground state from its origin, far from its equilibrium position, which can be detected by the time–dependent probe pulse at the bleach position.

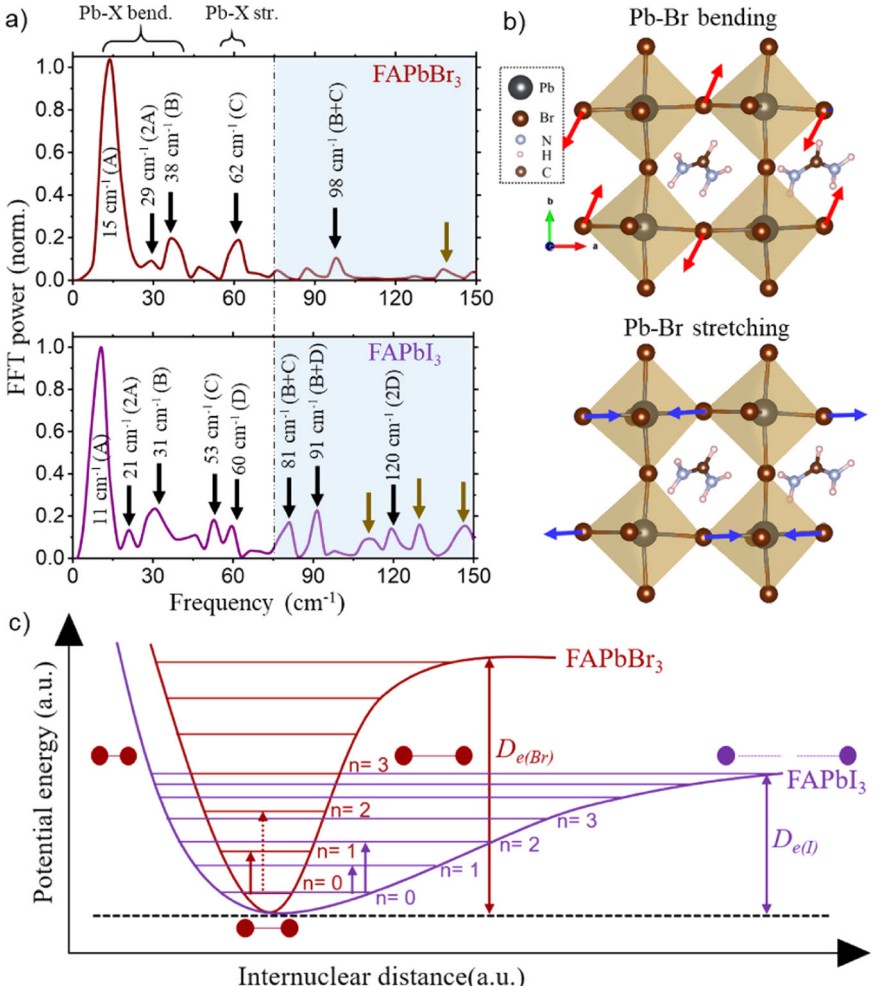

**Fig. 3 Anhamonicity in FAPbX₃ PNCs. a** The FFT power spectrum of the FAPbBr₃ PNCs at 510 nm and FAPbI₃ PNCs at 695 nm probe, respectively, computed over the first 10 ps time delay. The vertical dot-dashed line at 75 cm⁻¹ separates two regions of interest. The data shown are averaged over two datasets obtained from twice-repeated experimental observations. **b** Schematic of bending and stretching modes (red and blue arrows, respectively) in FAPbBr₃. **c** Schematic illustration of potential energy as a function of internuclear distance along with Pb-X bending (A-mode) coordinate, showing the extent of anharmonic character in FAPbBr₃ and FAPbI₃, as interpreted from the FFT analysis for the A-mode. $D_e$ stands for the bond dissociation energy.

the organic moiety present in the octahedral interstitial site of the PNCs.

Fig. 3a shows the fast Fourier transform (FFT) power spectrum of the FAPbBr₃ PNCs collected for the time trace at 510 nm, exhibiting a distribution of the vibrational features mostly below 150 cm⁻¹. The FFT analysis is also performed in the entire probe wavelength region (from 470 nm to 560 nm, see Supplementary Fig. 3) and it suggests that the coherent phonons have the strongest contribution only in the bleach region due to the ground state wavepackets. However, the coherent phonons can still contribute to the region, away from the bleach, due to the formation of the excited state wavepackets (see Supplementary Note 3)[18,38]. It is also important to note that the presence of the chirped probe can modify the relative weight of different phonon modes though correcting for the probe chirp does not allow to correct for the modification of the relative weight of the different phonon modes[39–42]. Although the time-domain data is analyzed after the chirp-correction, for the FFT power spectra construction we still used the raw data. Alongside the most prominent peak at 15 cm⁻¹ (A), two clear vibrational features are evident at 38 (B) and 62 (C) cm⁻¹ in the FFT power spectrum. Supplementary Table 1 summarizes all the coherently photoexcited low–frequency vibrational modes. Following the existing reports on

MAPbX₃[9–11,18,22], the lower frequency modes (15 and 38 cm⁻¹) are assigned to the bending of Pb–Br bonds, while the higher one (62 cm⁻¹) is associated with the stretching of Pb–Br bonds. Such an assignment of the vibrational modes can be further justified by our phonon calculations (see below). The bending (red arrows) and stretching (blue arrows) modes of FAPbBr₃ are schematically shown in Fig. 3b, as obtained from the first-principles calculations. If the low–frequency vibrations observed in the FFT analysis are indeed due to the lattice rearrangement of the inorganic Pb–Br framework (PbBr₆⁴⁻), it is expected that the observed vibrational frequencies would exhibit a strong halide dependence. To this end, we have synthesized FAPbBr₃₋ₓIₓ (where $0 < x < 3$) and FAPbI₃ PNCs by the ion–exchange method (see Supplementary Note 1) and performed the pump–probe measurements. The optical absorption (photoluminescence) spectra of the iodide–exchanged PNCs are found to be red-shifted to 585 nm (595 nm) and 707 nm (715 nm) for FAPbBr₃₋ₓIₓ and FAPbI₃ PNCs, respectively (Supplementary Fig. 4). The corresponding differential absorption spectra of the pump–probe measurements upon an above-bandgap non–resonant photoexcitation are shown in Supplementary Fig. 5. The strong phase–space filling signals for FAPbBr₃₋ₓIₓ and FAPbI₃ PNCs are observed at 588 nm and 707 nm, respectively. Similar to the pure bromide PNCs, on top of the

hot–carrier dynamics, the temporal traces of the FAPbBr$_{3−x}$I$_x$ and FAPbI$_3$ PNCs reveal a time-domain periodic oscillation due to the formation of the coherent phonons (Supplementary Fig. 6). To map out a more quantitative picture of the coherent phonons, FFT of the temporal traces have been carried out and is presented in Fig. 3a (bottom panel) (and Supplementary Fig. 7, Fig. 8). The most intense vibrational peaks appear at 12 cm$^{-1}$ and 11 cm$^{-1}$ for FAPbBr$_{3−x}$I$_x$ and FAPbI$_3$ PNCs, respectively, which correspond to the strongest vibrational feature observed at 15 cm$^{-1}$ (A) due to Pb–Br bending in the pure bromide PNCs. Therefore, these observations further confirm that the vibrational coherences are due to the structural rearrangement of the inorganic Pb–X framework (where X = halide), and the corresponding vibrational frequencies clearly show halide dependence, as summarized in Supplementary Table 1. The redshift of the vibrational frequency in FAPbBr$_{3−x}$I$_x$ and FAPbI$_3$ PNCs as compared to pure bromide PNCs can be explained by considering a harmonic oscillator approximation[43], based on the mass of the PbX$_6^{4-}$ unit.

Furthermore, along with the red-shift in the observed frequencies, it is also evident from the experimental FFT power spectra that a few more additional modes appear, especially in relatively higher frequency region (>75 cm$^{-1}$), for FAPbI$_3$ PNCs as compared to FAPbBr$_3$ PNCs (shaded area in Fig. 3a and Supplementary Fig. 7). Most of the observed additional vibrational modes can be well described by the higher harmonics (overtones or combination bands) of the fundamental modes (see Fig. 3a and Supplementary Table 1). For example, the higher harmonic frequency (first overtone) of the dominant Pb–I mode at 11 cm$^{-1}$ (A) in FAPbI$_3$ PNCs can be clearly observed at 21 cm$^{-1}$ (2 A). In addition, the modes that appear at 81 and 91 cm$^{-1}$ in FAPbI$_3$ PNCs can also be assigned to the combination bands, labeled as B + C and B + D, respectively. Similarly, in the case of the FAPbBr$_3$ PNCs, the first overtone of the corresponding fundamental mode (15 cm$^{-1}$) is also observed (c.a. at 29 cm$^{-1}$), albeit with reduced intensity. Previously, such higher harmonic modes were observed both experimentally and theoretically in semimetals such as Bi and Sb[44–46]. These modes were assigned to the direct coupling between the phonons (i.e., anharmonic

coupling), indicating higher anharmonicity of the lattice. Similarly, in this study, the appearance of the higher harmonics in FAPbI$_3$ PNCs may likely due to the more anharmonic character of PbI$_6^{4-}$ compared to PbBr$_6^{4-}$ because of the strong coupling within their fundamental phonon modes. Moreover, the combination bands that appear in the FFT spectra qualitatively explain the direct interaction between different phonons. We further note that some of the relatively high-frequency modes (>100 cm$^{-1}$) cannot be described by the higher harmonics of the fundamental modes in FAPbX$_3$ PNCs (shown with dark yellow arrows in Fig. 3a and Supplementary Fig. 7). These modes can be attributed to the librational motion associated with the FA molecule. Earlier, the appearance of such high frequency (>100 cm$^{-1}$) modes in MA–based perovskites were attributed to the MA's librational modes[47–49]. Interestingly, it is noticed that the contribution of the FA-librational motion increases in FAPbI$_3$ PNCs as compared to FAPbBr$_3$ PNCs, further supporting the anharmonicity model described in this work. It is worth mentioning that the broad vibrational modes in the case of mixed halide PNC (Supplementary Fig. 7) arise possibly due to the FA molecule facing an anisotropic interaction with the inorganic cage as compared to the case of pure halide PNCs. This is because the mixing of halide ions leads to inhomogeneity in the local crystal structure and reduces structural symmetry, resulting in the coupling of cation motion. Thus a more complex local lattice dynamics is expected[50,51]. In addition, we note that some of the higher harmonics and combination bands appear with similar intensity as the fundamental ones, the origin of which is unclear at present. This can be associated with different local dipole moments faced by the FA molecule in different halide environments. Although similar features were observed in bismuth metal[46], understanding the actual origin of such phenomenon in complex organic-inorganic hybrid perovskites needs further investigation. Fig. 3c illustrates the schematic representation of the corresponding potential energy surface of FAPbI$_3$ and FAPbBr$_3$ for the strongest A-mode (see the Discussion). As depicted in Fig. 3c, the higher anharmonic character in PbI$_6^{4-}$ break the harmonic selection rule (i.e., Δn =

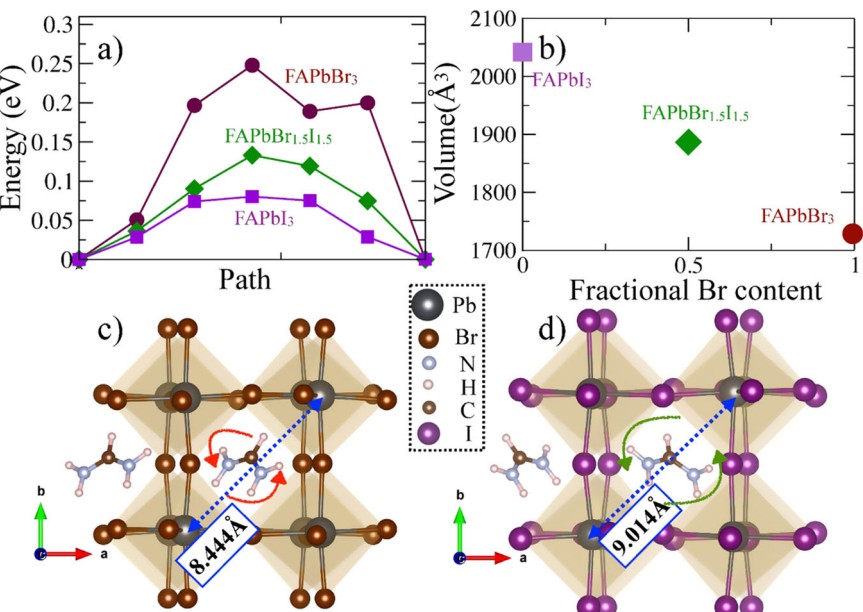

**Fig. 4 Role of the central organic moiety in the observed phonon dynamics. a** Calculated minimum-energy path for the rotation of FA moiety between two stable symmetrically equivalent configurations. The results are shown for FAPbBr$_3$ (maroon circle), FAPbBr$_{1.5}$I$_{1.5}$ (green diamond), and FAPbI$_3$ (purple square). **b** Change in volume of the 2 × 2 × 2 supercell (used for our DFT calculation) as a function of Br fraction. **c**, **d** show the relaxed structures of FAPbBr$_3$ and FAPbI$_3$, respectively. The numbers indicate the framework cage size.

±1) and allows additional transitions (Δn = ±2, …), which give rise to higher harmonics in the case of FAPbI₃. Although lattice anharmonicity in perovskites has been predicted before[9,52,53], to the best of our knowledge, this is the first experimental observation of an anharmonic character and its dependence on the nature of halide ion in hybrid halide perovskites. This is a significant advancement in the fundamental understanding of halide perovskite properties, including stability, conductivity, and polaronic character.

**Theoretical insights of observed lattice anharmonicity**. There are two possible reasons for the observed changes in the anharmonicity of the inorganic framework vibrations. First, the anharmonicity can increase due to changes in the ion interactions within the framework as Br is replaced with I. Second, the anharmonicity can be influenced by the interaction between the framework and the organic molecules. To understand how the latter is affected by the halide substitution, we have calculated the energy barrier associated with the rotation of FA moiety in bromide and iodide hybrid perovskites using all-electron DFT with Perdew-Burke-Ernzerhof (PBE) exchange-correlation functional[54] and Tkatchenko-Scheffler van der Waals interaction correction[55]. We have obtained the minimum energy paths associated with the FA moiety's rotations with the string method[56]. The corresponding potential energy surfaces are depicted in Fig. 4a. Note that the spin-orbit coupling (SOC) interactions do not affect our findings (see "Methods" section and SI for details). As can be seen in Supplementary Fig. 9, the SOC interactions do not modify the structural hierarchies, and hence do not alter our conclusions related to the rotational energy barriers or phonons. To find the most stable orientation of the FA moiety, we have performed several structural relaxations with different initial orientations of the FA cation inside the inorganic cage. As can be seen from Fig. 4a, the rotational energy barrier for FA between two stable symmetrically equivalent configurations inside the octahedral interstitial cage in FAPbBr₃ is three times higher than in FAPbI₃ (0.25 eV vs. 0.08 eV). These observations suggest that there is a stronger interaction between FA and PbBr₆⁴⁻ sublattice as compared to PbI₆⁴⁻. This is explained by the lower equilibrium cell volume of FAPbBr₃ compared to FAPbI₃, as shown in Fig. 4b. According to previous reports, FA prefers to orient along (001) direction in FAPbBr₃[20]. However, our supercell calculations suggest that the FA molecules in different cages prefer to align themselves in different orientations (see Supplementary Fig. 10). The orientation difference is again more pronounced for FAPbI₃ (Fig. 4c, d). Nonetheless, due to the Pm3m symmetry, no octahedral rotation or tilts are present in our studied systems to entangle the PbX₆-FA dynamics[20]. To further elucidate the effects of halide substitution on the rotational barrier faced by the FA molecules in FAPbBr₃₋ₓIₓ, we have constrained the FAPbI₃ lattice parameters to mimic that of FAPbBr₃. The rotational barrier of FA moiety in this constrained lattice (~0.252 eV) is similar to FAPbBr₃ (~0.25 eV) and is much higher than in FAPbI₃ (0.08 eV) (see Supplementary Fig. 11). This observation clearly indicates the significant dependence of the rotational energy barrier on the lattice expansion caused by iodine substitution.

## Discussion

The theoretical calculations show that the substitution of Br with I in FAPbBr₃ promotes the decoupling of the inorganic framework from the FA molecule. If the anharmonic coupling between the framework-associated phonons increases because of this decoupling, it might influence the observed energy transfer into the combination bands and overtones. However, we still need to separate this effect from the changes within the inorganic framework due to the substitution. To do this, we have performed a

similar experimental study on methyl ammonium lead bromide (MAPbBr₃) PNCs. Note that the size of the central organic moiety MA is much smaller compared to FA. The oscillations in the temporal dynamics (due to the formation of the coherent phonons) and the corresponding FFT spectrum are depicted in Supplementary Figs. 12 and 13, respectively. The observed frequency spectrum for MAPbBr₃ PNCs is closely comparable to the spectra previously reported for a bulk film of MAPbX₃ (X = Br and I)[18,31]. Interestingly, we have observed higher harmonics in MAPbBr₃ PNCs that are absent in the case of FAPbBr₃ PNCs (see Supplementary Fig. 13). Furthermore, we have calculated the rotational barrier of the MA molecule in MAPbBr₃. Its comparison with the rotational barrier of the FA moiety in FAPbBr₃ is presented in Supplementary Fig. 14. Despite having a similar halide inorganic cage (PbBr₆⁴⁻), the rotational energy barrier of the MA molecule is found to be significantly lower as compared to the FA molecule. This indicates that the inorganic framework in the hybrid perovskites is intrinsically anharmonic, but the anharmonicity is weakened by the interaction with the organic molecule. This observation confirms that the energy transfer between the framework phonons is indeed influenced by the interaction between the framework and the organic molecules, and not only by halide nature.

To confirm the assignment of experimentally observed phonon modes, we have additionally calculated the phonon eigenvectors and frequencies for FAPbBr₃ and FAPbI₃ systems within the harmonic approximation. The calculations suggest that the lower frequencies (<50 cm⁻¹) correspond to the bending of Pb-X bonds and the higher ones (up to 80 cm⁻¹) are mainly associated with the stretching of Pb-X bonds. Importantly, the theoretically calculated vibrational spectra are in good agreement with experimentally observed frequencies (see Supplementary Fig. 15). Note that our phonon modes are calculated at 0 K, whereas the experiments reported herein have been carried out at room temperature. Nevertheless, in accordance with the experimental findings, we find a prominent red-shift in the iodide perovskite. By further visually inspecting the phonon eigenvectors[20], we find that the FA moieties stop contributing to the inorganic cage vibrations in the FAPbI₃ system at a lower frequency (~120 cm⁻¹) compared to FAPbBr₃ (~188 cm⁻¹). This is consistent with the predicted weakening of the framework-molecule interaction in FAPbI₃.

For quantitative understanding of the observed anharmonicity, we have further estimated the anharmonicity constant along the strongest Pb-X bending coordinate for different PNCs. We think it is quite reasonable to consider only the strongest Pb-X bending coordinate for the anharmonicity calculation. This is because the ground-to-excited state displacement of the potential minima is maximized along the Pb-X bending at 14.7 cm⁻¹ (10.9 cm⁻¹) for FAPbBr₃ (FAPbI₃), and thus predominantly responsible for the excited state deformation (see Supplementary Fig. 16 and Note 3 for details). The anharmonicity constant $\chi_e$ can be estimated using Morse potential:

$$E_n = \omega_e\left(n + \frac{1}{2}\right) - \chi_e\omega_e\left(n + \frac{1}{2}\right)^2 \qquad (2)$$

Here, $E_n$ is the energy of the $n^{th}$ state. The fundamental vibrational frequency corresponds to the transition between $n = 0$ and $n = 1$, while the first overtone corresponds to the transition between $n = 0$ and $n = 2$. As these two transitions along with the strongest Pb–Br bending occur at 14.7 cm⁻¹ and 29.1 cm⁻¹ for FAPbBr₃, solving Eq. (2) yields $\chi_{e,Pb-Br}$~0.01. Similarly, the anharmonicity constant for FAPbBr₃₋ₓIₓ and FAPbI₃ is estimated to be $\chi_{e,Br-Pb-I}$~0.02 and $\chi_{e,Pb-I}$~0.04, respectively. The anharmonicity constants obtained from the experimental vibrational

modes clearly suggest that the anharmonicity in FAPbBr$_3$ PNCs is weakened by a factor of 4 due to interaction of the Pb-Br sublattice with the FA molecule, as compared to FAPbI$_3$ PNCs. At this point, we would like to clarify that although we observe a positive anharmonicity constant for the A-mode, however, in principle, the anharmonicity can be of both signs, different for different modes. This was demonstrated for both molecules and solids[57,58]. Therefore, in this study, the anharmonicity model depicted in Fig. 3c is limited to the A-mode only.

The dependence of the interaction between the framework and the organic molecules on the halide nature can explain several fundamental properties of hybrid halide perovskites. The reduced interaction of the FA cation with the Pb–I sublattice leads to a more fragile Pb–I sublattice due to its free vibration and therefore, can be correlated with the lower stability of iodide (FAPbI$_3$) compared to bromide (FAPbBr$_3$) perovskites, as observed experimentally[59]. To further quantify the stability of bromide perovskite over the iodide one, we have also calculated the formation energies of the perovskites using the following formula:

$$E_{Form} = E_{FAPbX_3} - E_{FAX} - E_{PbX_2} \qquad (3)$$

Here, $E_{Form}$ is the formation energy of the halide perovskite, and $E_{FAPbX_3}$, $E_{FAX}$, $E_{PbX_2}$ are the total energies of the FAPbX$_3$, FAX, and PbX$_2$ respectively. We found that the bromide perovskite with $E_{Form} = -0.75$ eV is more stable with respect to its iodide counterpart having $E_{Form} = -0.22$ eV (i.e., by 0.53 eV per formula unit). Effects of halide nature on the interaction between the framework and the organic molecules can also explain the lower charge–carrier conductivity of FAPbBr$_3$ compared to FAPbI$_3$, as observed by terahertz photoconductivity measurements[14], where the enhanced interaction of the FA dipoles with the PbBr$_6^{4-}$ framework results in an increase of carrier scattering. Furthermore, this can explain the higher inhomogeneous broadening observed in the optical transitions in FAPbBr$_3$[10], due to the fact that the enhanced interaction between the FA moiety and the Pb–Br sublattice in FAPbBr$_3$ leads to higher inherent disorder as compared to FAPbI$_3$.

In summary, we have elucidated the halide-dependence of the excited state vibrational coherences of FAPbX$_3$ (X = Br and I) PNCs by femtosecond pump–probe spectroscopy. Above-bandgap photoexcitation of the colloidal PNCs results in coherent vibrational wave–packets that can be resolved into different vibrational modes of their inorganic cage. For FAPbI$_3$ PNCs, along with the red–shift in the frequencies, additional vibrational modes get activated at frequencies >75 cm$^{-1}$, which can be well described by the higher harmonics of their fundamental frequencies. This indicates the potential energy surface is more anharmonic in nature in the case of FAPbI$_3$ PNCs as compared to FAPbBr$_3$ PNCs. The red–shift is caused by the higher inertia for lattice rearrangement of the PbI$_6^{4-}$ sublattice as compared to relatively lighter PbBr$_6^{4-}$. However, due to the larger cavity size (where the FA$^+$ sits) in FAPbI$_3$, the interaction of the inorganic framework with the FA moiety is weaker than in FAPbBr$_3$. This weakening allows for the intrinsic anharmonicity of the inorganic framework to be revealed, facilitating the energy transfer into overtone and combination bands. Importantly, our control experiment in MAPbBr$_3$ revealed the energy transfer between the framework phonons due to the intrinsic anharmonicity of the lead–halide framework is indeed influenced by the interaction between the framework and the organic molecules, as well as by halide nature. These results provide the fundamental insight into the lattice rearrangement of the inorganic cage, as well as the rotational energy barrier of the central organic moiety in organic–inorganic halide PNCs that can address several fundamental properties of such PNCs.

## Methods

### Experimental

*Materials.* Formamidinium (FA) acetate, methyl ammonium bromide (MABr), lead bromide (PbBr$_2$), lead iodide (PbI$_2$), octadecene, oleic acid, oleylamine, hexane, toluene, DMF were obtained from Sigma-Aldrich and used without further purification.

*Synthesis of PNCs.* FA acetate (1.88 mmol) along with octadecene (9 mL) and oleic acid (6 mL) were loaded into a three-neck flask and degassed for 1 h at room temperature. The mixture was then heated to 100 °C under nitrogen until complete dissolution before it cooled down to room temperature. Thus prepared FA-oleate was kept for further use.

To prepare the FAPbBr$_3$ PNCs, PbBr$_2$ (0.133 mmol) salt along with 0.5 mL oleylamine, 1 mL oleic acid, and 5 mL octadecene were taken in another three-neck flask. The reaction mixture was then subjected to dry under vacuum at 100 °C for about half an hour. The temperature of the reaction mixture was increased to 150 °C under nitrogen and 2.5 mL of FA-oleate stock solution was injected into the reaction flask. The reaction mixture was quenched immediately by a water-ice bath. The crude solution was centrifuged at 4629 rcf for 5 min and the supernatant was discarded. The precipitate was dissolved in hexane and stored for future experiments.

*Steady-state measurements.* UV-vis absorption and PL spectra of FAPbBr$_3$, FAPbBr$_{3-x}$I$_x$, and FAPbI$_3$ PNCs were performed using Cary 50 UV−vis spectrophotometer (Varian) and Cary Eclipse (Varian) spectrometer respectively, after dispersing the PNCs in hexane.

*Pump–probe spectroscopy.* Vibrational coherence measurement has been performed through the ultrafast pump–probe spectroscopy in a custom–built transient absorption spectrometer (Newport TAS). The pump–probe spectroscopy is based on a multi-pass Ti: Sapphire laser system (Libra, Coherent Inc.). The Ti: Sapphire amplifier system produces ~100 fs laser pulses at a 1 kHz repetition rate having a central wavelength of 800 nm. Pump pulses at 400 nm were generated via frequency doubling of the original pulse (800 nm) using a β-barium borate (BBO) crystal and were then focused onto the sample. To generate the visible probe, the other part of the initial pulses (~1 μJ) at 800 nm were guided through a motorized CaF$_2$ crystal and further split to probe and reference beams. The pump and probe pulses were overlapped spatially and temporally in the sample. In addition, an optical delay line, controlled by a motorized delay stage, was positioned in the probe beam path to offer a time delay between pump and probe beams. A 0.5 kHz optical chopper was also placed on the pump arm to read out the changes in the probe transmission for each pump–probe cycle. The pump and probe spectrum and the energy stability of the 400 nm pump and the white light probe pulse are provided in Supplementary Figs. 17, 18, and Note 4, 5. The average number of excitons were kept low enough («1) to avoid phenomenon due to multiexciton generation for all measurements. All the experiments were performed using the ~100 fs laser pulse in 50-fs steps and by keeping the excitation photon density ~3.5 × 10$^{13}$ cm$^{-2}$. The samples were kept in a 2-mm cuvette and the colloidal solution was continuously stirring using a magnetic stirrer to avoid sample bleaching during the experiment. Data analysis and plotting were carried out using Matlab, Lab-View, and Origin software. The FFT was performed on the residual time trace, obtained after exponential subtraction of a specific time trace, in Origin software. The FFT spectral resolution is ~3.33 cm$^{-1}$ for the ~10 ps temporal window in this work[38]. The absorption and PL data, the TA spectrum, and the FFT spectra are plotted after applying *Spline* in Origin software for a smoother representation.

*Theoretical.* First-principles calculations within the framework of density functional theory are carried out using all-electron full-potential electronic-structure package FHI-aims[60] using a numeric atom-centered basis set. Electronic exchange and correlation are treated with Perdew-Burke-Ernzerhof (PBE)[54] generalized gradient approximation (GGA). Long-range pairwise van der Waals interaction correction is calculated using Tkatchenko-Scheffler (TS) approach[55]. It has been demonstrated that TS reduces the unit-cell volume over-estimations by PBE from 6.5% to 0.9% in FAPbBr$_3$[20] while considering many–body dispersion reduces the same to only 1.5%. The numeric settings are chosen so that a force convergence of less than 10$^{-2}$ eV/Å is achieved. All the atomic positions and the lattice vectors are allowed to relax fully. We have taken the initial symmetry from experimental XRD results[33] for FAPbBr$_3$, which is pseudo-cubic with space group Pm3m. Following that, the initial symmetry of the substitutional structures has been kept the same as FAPbBr$_3$, while allowing a total relaxation of atomic positions and lattice in all calculations. A dense k-point mesh of 6 × 6 × 6 points has been used for all the calculations. To find the most energetically favored geometrical configuration for FAPbBr$_x$I$_{3-x}$ (x = 1.5), we first have identified the most favorable site for I substitution by scanning over all the halide (X) sites in a 2 × 2 × 2 supercell. Next, by keeping that site fixed (i.e., substituted with I), we again have scanned over the remaining sites to find the next favorable lattice site for I substitution and so on. We have obtained the minimum energy paths associated with the FA moiety's rotations with the string method[56]. Climbing image feature[61], as implemented in AIMSCHAIN[60], is used with six images, ensuring a tight convergence of force to

0.05 eV/Å. To understand if the SOC interactions play any significant role in our analysis, we have calculated the energies of FAPbBr$_3$ unit cell as a function of unit cell volume with and without SOC[62] interactions (see Supplementary Fig. 9). The phonons are calculated using the finite displacement method with harmonic approximation, as implemented in PHONOPY code[63]. While we have calculated the rotational barrier for the primitive unit cell, enforcing periodic boundary conditions on the FA moieties, full structural relaxation of $2 \times 2 \times 2$ supercell has been carried out before phonon calculations.

## Data availability

Data is available from the corresponding author upon reasonable request. The data files are also available in the NOMAD data repository (https://doi.org/10.17172/NOMAD/2021.03.23-1).

## Code availability

We have used two codes for our calculations: FHI-aims (https://aimsclub.fhi-berlin.mpg.de/) and PHONOPY (https://phonopy.github.io/phonopy/). PHONOPY is an open-access code; for details of the FHI-aims license, please visit the website.

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

## Acknowledgements

We acknowledge financial support by the Bavarian State Ministry of Science, Research, and Arts through the grant "Solar Technologies go Hybrid (SolTech)", the Deutsche Forschungsgemeinschaft (DFG, German Research Foundation) under Germanys Excellence Strategy—EXC 2089/1-390776260, the Alexander von Humboldt Foundation (T.D. and A.D.), the German-Israeli Foundation for Scientific Research and Development (GIF, Project I-1512-401.10/2019), the European Union's Horizon 2020 research and innovation program under the Marie Sklodowska-Curie grant agreement No. 839042 (H.H.). S.V.L. was supported by RFBR and INSF, project number 20-53-56065. We thank Boaz Pokroy (Technion, Israel) for fruitful discussions.

## Author contributions

T.D. and J.F. originated the project concept. T.D. performed all the experiments. T.D. initiated and coordinated with D.S. and S.V.L. for theoretical calculations. D.S., Z.K.H., and S.V.L. performed the theoretical calculations. H.H., A.D., and T.D. carried out all the synthesis and characterization. All authors contributed towards the interpretation of the results and the discussion of the outline of the manuscript. T.D., A.D., L.P., and J.F. discussed the final figures and outline of the experimental part of the manuscript. D.S., T. D., and S.V.L. discussed the final figures and outline of the theoretical part of the manuscript. Finally, T.D. and D.S. wrote the manuscript, with input from all the authors.

## Funding

## Competing interests

The authors declare no competing interests.
