## [Peer Review File · Nature Communications]

REVIEWER COMMENTS

Reviewer #1 (Remarks to the Author):

Debnath et al apply transient absorption spectroscopy to a series of FAPbX perovskite nanocrystals. By exciting coherent oscillations, they are able to extract time-domain vibrational fingerprints of each material. Variation of the halide component leads to subtle and systematic changes in the vibrational spectrum, which they relate to increasing anharmonicity of the lattice with increasing iodide content. This conclusion is supported by a computational investigation of the strength of interaction between the FA moiety and surrounding lattice as the halide content is changed. The experiments are well conceived, the conclusion is reasonable, and the findings are of interest to the community. However, I have some concerns about how strongly the experimental data shows what the authors claim, and the data analysis should be made more robust. If these issues can be addressed I believe this work has potential for publication in Nature Communications. My detailed comments follow.

- 1) What determines the detection threshold for vibrational peaks in Figure 3a and the corresponding SI data? I can find no reference to the methodology, and the data appear to be inconsistently treated throughout (threshold is different in every experiment, in the case of MAPbBr₃ drastically so).
- 2) The prominent lowest-frequency peak is clear in all experiments, and the assignments to B and C in the FAPbBr₃ data seem reasonable. But the peak attributed to B+C is scarcely stronger than the unlabelled peak ~30 cm⁻¹. I note the latter peak is assigned as 2A in the SI; why was it omitted from the main?
- 3) In the FAPbI₃ data, the dashed threshold appears slightly higher yet many more peaks are identified. However, three peaks that meet or exceed the threshold (~45 cm⁻¹, ~112 cm⁻¹, ~128 cm⁻¹) are not assigned. What are they?
- 4) In the FAPbBrI data in the SI, the dashed threshold is set to a different level again and every single peak is labelled. However, some of these are broad and ill-defined (e.g. C, B+D) despite the corresponding fundamental/combination bands (B, D, B+C) being sharply resolved. Is this reasonable?
- 5) In the FAPbBr₃ data, the overtone frequencies should appear at or less than integer multiples of the fundamental frequency, not greater than. Likewise, in the FAPbI₃ data the authors label numerous combination or overtone modes and draw important conclusions from them, but the reported frequencies are largely inconsistent with the anharmonicity model. E.g. the 2D mode appears at more than 2x the frequency of D, and the B+C+D mode appears at a higher frequency than the sum of B, C and D modes, when they should in fact appear at lower frequencies. Are the frequencies not known as precisely as the authors indicate?
- 6) Can the authors explain why in the PbI₃ and PbBrI data many of the higher harmonics and combination bands appear with essentially the same intensity as the fundamental modes?
- 7) The authors argue that the presence of stronger overtones like 2A in the PbI₃ data is consistent with greater anharmonicity in that lattice. But the tentatively assigned 2A band in the PbBr₃ seems only slightly less intense, and the 2A band in the intermediate PbBrI data in the SI shows a markedly more pronounced 2A than either pure sample. Can this be explained?
- 8) The sum of these issues causes some concern about the peak assignments and resulting interpretation. Are they reliable? How do we know what is real here versus noise? If there is some degree of uncertainty in the peak frequencies, how strongly should we interpret the 2-4cm⁻¹ shift of the A mode?
- 9) The cartoon in Figure 2e indicates that wavepackets are excited and observed in the excited potential energy surface. However, Liebel (ref 37) and colleagues have extensively demonstrated how the principal vibrational contribution in such experiments arises from ground-state vibrational activity. The present study does not utilize additional 'population control' pulses and monitors the peak of the ground-state bleach, where ground-state vibrational coherence would be expected to make the greatest contribution. Thus, what evidence is there that the observed oscillations are due to wavepacket motion in the excited potential energy surface? It seems to me that it doesn't matter much for the authors' interpretation whether ground- or excited-state vibrational activity is detected, but the distinction needs to be addressed.

Reviewer #2 (Remarks to the Author):

Referee report for the manuscript "Coherent Vibrational Dynamics Reveals Lattice Anharmonicity in Organic-inorganic Halide Perovskite Nanocrystals" by T. Debnath et al., submitted to Nature Communications. Comments to the authors.

The topic of the manuscript by Debnath et al. is lead halide perovskite studied by time domain vibrational spectroscopy, an intriguing and timely subject of research.

The main idea of the manuscript is to coherently excite vibrational wave-packets by a resonant 400 nm femtosecond pump pulse, and then measure the phonon properties by recording in the time-domain the transmission of a temporally delayed probe pulse. By scanning the delay ΔT between the pump and the probe pulses and by Fourier transforming over ΔT , the authors can reconstruct the Raman spectrum of the sample under investigation.

The technique is presented and then applied to compare the phonon properties of FAPbI₃, FAPbBr₃ and MAPbBr₃. In particular, in FAPbI₃ the authors report several high intensity modes at frequencies higher than 75 cm⁻¹, which are assigned to higher harmonics of the inorganic cage fundamental modes, pointing to a strong anharmonicity in the Pb-I framework. The experimental observations are supported by all-electron DFT calculations, used to calculate the energy barrier associated with the rotation of FA moiety in bromide and iodide hybrid perovskites, which suggests a stronger interaction between the FA and PbBr sublattice with respect to PbI.

These observations are used to suggest an improved stability of the Br-based hybrid perovskites over the I-based ones.

While I find this study to be interesting, there are several key points that remain unclear or confusing and which the authors should address in a revised version of the manuscript. In addition, I believe that a stronger and more quantitative connection between experimental results and the sample anharmonic properties would be appropriate for a publication in Nature Communications. For these reasons I suggest a major revision.

My opinion is based on the following points.

- 1) By looking at Fig. 2, it seems that there is a noisy periodic modulation of the time-dependent transient differential absorption also for wavelengths red-shifted with respect to bleach maximum (figure attached). In such a sample transparent region, I would not expect to see oscillations in the time-domain (or at least, they should be less intense by several orders of magnitude). The presence of such a noise can generate artefact in the frequency domain Raman

spectrum, compromising the interpretation of all the manuscript. The authors should comment on that and report the Raman spectra extracted from the Fourier transformation over all the monitored probe wavelengths.

- 2) It looks that the authors have not been able to estimate and quantify from the experimental data the value of the anharmonicities in the different samples. I believe that the lack of such an estimate greatly diminishes the value of the manuscript. Furthermore, it is not clear how much the experimentally detected effect is quantitatively linked to improved stability of the Br-based hybrid perovskites.
- 3) Page 6: the authors claim that “To our surprise, the negative differential absorption signal (bleaching) shows a periodic oscillation in the time domain”. It is not a surprise detecting oscillation in the time-domain in these samples, they have been already reported in several papers: Refs. 18, 30 for example, but also “Free Carrier Emergence and Onset of Electron–Phonon Coupling in Methylammonium Lead Halide Perovskite Films” *J. Am. Chem. Soc.* 2017, 139, 50, 18262-18270, which is actually the first report of Raman oscillations in the time domain measured in lead halide perovskite films. All these references should be appropriately cited at this point of the manuscript.
- 4) In the manuscript, the authors refer to the pump as an off-resonant pulse: I believe that this is misleading, being the frequency of the pump pulse above the bandgap, it is strongly absorbed and can generate photo-carriers.
- 5) There is no discussion here of whether photodamage is an issue (and, in fact, it is very hard to tell since it occurs immediately upon photoirradiation). The experiment has been performed with a high frequency 400 nm pump pulse, i.e. in a spectral range where the sample absorbance is much higher with respect to the bandgap. The presence of photodamage effect can compromise the interpretation and should be accurately discussed.
- 6) The description of the experimental setup lacks several essential pieces of information: the spectrum, the time duration, the fluence, the stability of the

400 nm pump pulse as well as of the probe pulse. Furthermore, the presence of a chirped probe can modify the relative weight of different phonon modes: such effects, with appropriate references to previously published works (doi.org/10.1039/B920356G: Phys. Chem. Chem. Phys., 2010,12, 2149-2163, 10.1021/acs.jpcllett.7b00559: J. Phys. Chem. Lett. 2017, 8, 8, 1920-1924, 10.1021/acs.jpcllett.6b03027: J. Phys. Chem. Lett. 2017, 8, 5, 966-974 and 10.1021/acs.jpcllett.9b03061: J. Phys. Chem. Lett. 2019, 10, 24, 7789-7796), should be commented.

- 7) The temporal sampling interval and the temporal sampling window are missing and should be reported in the manuscript. They are particularly critical since they determine the spectral resolution and frequency window extracted upon the conversion to the frequency domain by the Fourier transform.
- 8) Equation on page 6 should read as $e^{in\Delta\omega T}$, with $n=1, 2, \dots$ (terms $e^{i(n+1)\Delta\omega T}$, $e^{i(n+2)\Delta\omega T}$, ... are not necessary and redundant). Immediately after such Eq. " $\Delta\omega = h/T$ " should read as " $\Delta\omega = 2\pi/T$ ".

Reviewer #3 (Remarks to the Author):

In the manuscript "Coherent Vibrational Dynamics Reveals Lattice Anharmonicity in Organic inorganic Halide Perovskite Nanocrystals", Debnath and co-authors have used a synergy between experiments and ab-initio quantum mechanical simulations to investigate and elucidate the origin of anharmonicity in hybrid halides perovskites, in particular FAPbI₃ and FAPbBr₃.

Investigating the peculiarities and differences in this family of materials is interesting to a broad community as it can unlock new key functionalities and applications. The manuscript is clear and well written. The experiments are well described and reproducible. The calculation method is described in details.

I recommend the paper for publication after these comments below are addressed:

- 1) The authors have not included spin-orbit coupling interactions in their calculations. Would the SOC change their results?
- 2) What is the symmetry of the investigated structures? Are all with the same symmetry?
- 3) The authors have considered a specific rotation of the FA molecule. It might be that in experiments multiple rotations are present at the same time. Can molecular dynamics simulations help in clarify the role of rotation with respect to the changes in the lattice and thus of the interaction between the molecule and the inorganic framework?
- 4) The combined effects of the interaction between the inorganic framework and role of anions could be decoupled by applying strain to FAPbI₃/FAPbBr₃ and changing the volume. Can the authors comment on this and show some preliminary data?
- 5) In the methods, the authors write "To find out the most energetically favored geometrical configuration for FAPbBr_xI_{3-x} (x = 1.5), we first have identified 313 the most favourable site for I substitution by scanning over all the halide (X) sites in a 2x2x2 supercell...". It is not clear why they use FAPbBr_xI_{3-x} as this structure is not mentioned in the manuscript and SI
- 6) It would be useful if the authors collect all calculations in a database and make it available in online repositories.

Reply to the Reviewers' comments

Reviewer #1 (Remarks to the Author):

Debnath et al apply transient absorption spectroscopy to a series of FAPbX perovskite nanocrystals. By exciting coherent oscillations, they are able to extract time-domain vibrational fingerprints of each material. Variation of the halide component leads to subtle and systematic changes in the vibrational spectrum, which they relate to increasing anharmonicity of the lattice with increasing iodide content. This conclusion is supported by a computational investigation of the strength of interaction between the FA moiety and surrounding lattice as the halide content is changed. The experiments are well conceived, the conclusion is reasonable, and the findings are of interest to the community. However, I have some concerns about how strongly the experimental data shows what the authors claim, and the data analysis should be made more robust. If these issues can be addressed I believe this work has potential for publication in Nature Communications. My detailed comments follow.

We thank the reviewer for appreciating our work and for the constructive comments. We hope that our response herein, along with the accompanying revised manuscript, would clarify the reviewer's concerns on the data analysis (peaks assignment).

1) What determines the detection threshold for vibrational peaks in Figure 3a and the corresponding SI data? I can find no reference to the methodology, and the data appear to be inconsistently treated throughout (threshold is different in every experiment, in the case of MAPbBr₃ drastically so).

The threshold line was arbitrary and it was set to guide the eye for separating the vibrational peaks having sufficient intensity from the noise level. However, we agree that by doing this we may neglect contribution from some of the important low-intensity vibrational peaks. To make a more consistent treatment of the data, we have removed the arbitrary threshold in the revised manuscript (Figure 3a and in the SI data), and all the peaks having sufficient intensity are assigned to specific vibrational modes. By doing so, we are able to extract important vibrational contributions from the FA-librational modes (see comment 3, 4 and 5), which were previously neglected.

2) The prominent lowest-frequency peak is clear in all experiments, and the assignments to B and C in the FAPbBr₃ data seem reasonable. But the peak attributed to B+C is scarcely stronger than the unlabelled peak ~30 cm⁻¹. I note the latter peak is assigned as 2A in the SI; why was it omitted from the main?

We thank the reviewer for pointing this out. Sorry, we missed to label the peak $\sim 30\text{ cm}^{-1}$ although the assignment was mentioned in the main as “Similarly, in the case of the FAPbBr₃ PNCs, the first overtone of the corresponding fundamental mode (14 cm⁻¹) is also observed (c.a. at 28 cm⁻¹), albeit with reduced intensity” (Page 9). We have labelled this peak in Figure 3a of the revised manuscript (reproduced as Figure R1 below).

Figure R1. The FFT power spectrum of the FAPbBr₃ PNCs at 510 nm and FAPbI₃ PNCs at 695 nm probe, respectively, computed over the first 10 ps time delay.

3) In the FAPbI₃ data, the dashed threshold appears slightly higher yet many more peaks are identified. However, three peaks that meet or exceed the threshold ($\sim 45\text{ cm}^{-1}$, $\sim 112\text{ cm}^{-1}$, $\sim 128\text{ cm}^{-1}$) are not assigned. What are they?

After removing the arbitrary threshold limit, we are able to extract a few more vibrational peaks in the FAPbI₃ data, which are pointed out by the reviewer. The relatively high-frequency peaks at $\sim 111\text{ cm}^{-1}$, $\sim 130\text{ cm}^{-1}$, and 147 cm^{-1} (See Figure R1, dark-yellow arrows) can be attributed to the contribution of FA-librational motion, similar to MA-librational motion discussed in previously published literature (Ref. R1-R5). However, we are unable to assign the observed hump at $\sim 46\text{ cm}^{-1}$ at this point. We have included the new assignments in the revised manuscript along with suitable references as discussed here (page 9).

4) In the FAPbBrI data in the SI, the dashed threshold is set to a different level again and every single peak is labelled. However, some of these are broad and ill-defined (e.g. C, B+D) despite the corresponding fundamental/combination bands (B, D, B+C) being sharply resolved. Is this reasonable?

Indeed a few peaks appeared to be broad in FAPbBrI data; this is likely due to the anisotropic interactions of FA molecule with the inorganic cage made of mixed halides as compared to that of the case with a single halide (see comment 7). In addition, we have removed the dashed threshold for consistency and assigned the higher harmonic modes more reasonably (see Figure R2). For example, the broad mode appearing at 109 cm^{-1} is now identified as the first overtone of the broad fundamental stretching mode at 55 cm^{-1} (C). In addition, a couple of additional low-intense modes are identified at relatively higher frequency (129 cm^{-1} and 142 cm^{-1}), indicated by dark-yellow arrows, which can be attributed to the contribution of FA-librational motion (as discussed above). We have included these peak assignments and the corresponding discussion in the revised manuscript (page 9) and SI.

Figure R2. The FFT power spectrum of the FAPbBr_xI_{3-x} PNCs at 590 nm, computed over the first 10 ps time delay.

5) In the FAPbBr₃ data, the overtone frequencies should appear at or less than integer multiples of the fundamental frequency, not greater than. Likewise, in the FAPbI₃ data the authors label numerous combination or overtone modes and draw important conclusions from them, but the reported frequencies are largely inconsistent with the anharmonicity model. E.g. the 2D mode appears at more than 2x the frequency of D, and the B+C+D mode appears at a higher frequency than the sum of B, C and D modes, when they should in fact appear at lower frequencies. Are the frequencies not known as precisely as the authors indicate?

We thank the reviewer for pointing out the small variation in overtone frequencies. We have revisited the observed frequencies more precisely and found that D-mode appear at $\sim 59.7\text{ cm}^{-1}$ (and approximated to 60 cm^{-1}) while the 2D mode appears at $\sim 119.5\text{ cm}^{-1}$ (approximated to 120 cm^{-1}) in FAPbI₃. The previously assigned B+C+D mode at 147 cm^{-1} , however, is likely attributed to FA-librational motion (dark-yellow arrow in Figure R1), in line with previously published literature for MA-based perovskite (Ref. R1-R5). We have included these corrections in the revised manuscript (page 7, 9).

At this point, we would like to clarify that although the reviewer suggested the overtone frequencies should appear at or less than integer multiples of the fundamental frequency, however, we don't think this is true in general. The anharmonicity can be of both signs, different for different modes. In fact in molecules, the overtone of asymmetric stretching and bending can be of higher frequency because of anharmonicity (Ref. R6). This was demonstrated for solids as well (Ref. R6-R7). In the present study, however, we observed that

the overtone frequencies appear at or less than integer multiples of the fundamental frequency, see SI Table 1. We further calculated the anharmonicity constant for the strongest A-mode and found positive anharmonicity constant for all three samples (see comment 2 from Reviewer 2). Therefore, the anharmonicity model presented in figure 3c is mainly based on the A-mode (i.e. Pb-X bending). In the revised manuscript, we have clearly mentioned this (page 7, 13).

6) Can the authors explain why in the PbI₃ and PbBrI data many of the higher harmonics and combination bands appear with essentially the same intensity as the fundamental modes?

We thank the reviewer for this interesting observation. Its origin is unclear at present. Previously, a similar observation was found in bismuth metal where the higher harmonic mode appears with similar intensity as the fundamental mode (see Ref. R9). Although the origin of such observation may vary from system to system, it is reported that the intensity of a particular vibrational mode depends strongly on the associated dipole moment (Ref. R10, R11). In the present study, different halide environment faced by the FA molecule may result in an alternation of the local dipole moment, which can modify the intensity of the higher harmonic modes. This would be an interesting topic for future studies. We have included a short discussion regarding this remaining issue in the revised manuscript (page 9).

7) The authors argue that the presence of stronger overtones like 2A in the PbI₃ data is consistent with greater anharmonicity in that lattice. But the tentatively assigned 2A band in the PbBr₃ seems only slightly less intense, and the 2A band in the intermediate PbBrI data in the SI shows a markedly more pronounced 2A than either pure sample. Can this be explained?

This is also an interesting observation. In the case of mixed halide PNC, the FA molecule faces anisotropic interactions as compared to the case of pure halide PNCs. This is because the mixing of halide ions leads to inhomogeneity in the local structure and reduces structural symmetry, which may result in the coupling of cation motion (see R8-R10) and equilibrium distribution to more complex local lattice dynamics. We have included this discussion in the revised manuscript (page 9).

8) The sum of these issues causes some concern about the peak assignments and resulting interpretation. Are they reliable? How do we know what is real here versus noise? If there is some degree of uncertainty in the peak frequencies, how strongly should we interpret the 2-4cm⁻¹ shift of the A mode?

We understand the concerns of the reviewer regarding the peaks assignment and the noise peaks. Following Reviewers' suggestion, we have addressed all the concerns about the peak assignments and resulting interpretation more precisely (as discussed above) which makes the result more reliable and robust. To minimize the effect of noise, we addressed the modes having intensity up to two orders of magnitude lower than the strongest mode A (but not below).

Based on the following facts, we could clearly interpret the 2-4 cm⁻¹ shift of the A mode in different PNCs:

- i) The first overtone modes for FAPbBr₃, FAPbBr_xI_{3-x} and FAPbI₃ appeared at 29 cm⁻¹, 25 cm⁻¹ and 21 cm⁻¹, respectively. The 8 cm⁻¹ shift of the first overtone peak from bromide to iodide, however, cannot be explained by the small degree of uncertainty in the peak frequencies.
- ii) As mentioned in the main text, “*The redshift of the vibrational frequency in FAPbBr_{3-x}I_x and FAPbI₃ PNCs as compared to pure bromide PNCs can be explained by considering a harmonic oscillator approximation,³⁸ based on the mass of the PbX₆⁴⁻ unit.*” Considering the mass of the unit cells PbBr₆⁴⁻ and PbI₆⁴⁻, we can predict the frequency ratio, using the harmonic approximation ($f=1/2\pi \sqrt{(k/m)}$) and it turns out to be $f_{\text{Pb-Br}}/f_{\text{Pb-I}} \sim 1.2$. This indeed supports the 2-4 cm⁻¹ shift of the A mode observed in different PNCs.
- iii) Being the strongest mode, it is easy and more reliable to detect the peak frequencies of A mode in different halide PNCs.
- iv) Finally, the FFT data shown in Figure 3a are averaged over two datasets (having quite similar frequency values of the A-mode) obtained from twice-repeated experimental observations (on different days) i.e. peak frequency of A mode remains unaltered on day to day basis for FAPbX₃.

9) The cartoon in Figure 2e indicates that wavepackets are excited and observed in the excited potential energy surface. However, Liebel (ref 37) and colleagues have extensively demonstrated how the principal vibrational contribution in such experiments arises from ground-state vibrational activity. The present study does not utilize additional “population control” pulses and monitors the peak of the ground-state bleach, where ground-state vibrational coherence would be expected to make the greatest contribution. Thus, what evidence is there that the observed oscillations are due to wavepacket motion in the excited potential energy surface? It seems to me that it doesn’t matter much for the authors’ interpretation whether ground- or excited-state vibrational activity is detected, but the distinction needs to be addressed.

We agree with the reviewer that monitoring ground-state bleach provides information mostly about ground-state vibrational coherence. In the present study, indeed, the principal vibrational activity arises from the ground-state vibrational wave packets (Figure R3). We have corrected this in the revised manuscript (along with Figure 2e).

Figure R3. Illustration of a ground-state vibrational wave packet motion along a particular phonon displacement coordinate of PbBr_6^{4-} octahedral, generated by interaction of the pump pulse with the PNCs.

References (Reviewer 1):

- R1. Quarti, C., Mosconi, E. & De Angelis, F. Structural and Electronic Properties of Organohalide Hybrid Perovskites from Ab initio Molecular Dynamics. *Phys. Chem. Chem. Phys.* **17** (14), 9394–9409 (2015).
- R2. Neukirch, A. J., Nie, W., Blancon, J.-C., Appavoo, K., Tsai, H., Sfeir, M. Y., Katan, C., Pedesseau, L., Even, J., Crochet J. J. et al. Polaron Stabilization by Cooperative Lattice Distortion and Cation Rotations in Hybrid Perovskite Materials. *Nano. Lett.* **16**, 3809–3816 (2016).
- R3. Niemann, R. G., Kontos, A. G., Palles, D., Kamitsos, E. I., Kaltzoglou, A., Brivio, F., Falaras, P. & Cameron, P. J. Halogen Effects on Ordering and Bonding of CH_3NH_3^+ in $\text{CH}_3\text{NH}_3\text{PbX}_3$ (X = Cl, Br, I) Hybrid Perovskites: A Vibrational Spectroscopic Study. *J. Phys. Chem. C* **120**, 2509–2519 (2016).
- R4. Ghosh, T., Aharon, S., Etgar, L. & Ruhman, S. Free Carrier Emergence and Onset of Electron–Phonon Coupling in Methylammonium Lead Halide Perovskite Films. *J. Am. Chem. Soc.* **139**, 18262–18270 (2017).
- R5. Park, M., Neukirch, A. J., Reyes-Lillo, S. E., Lai, M., Ellis, S. R., Dietze, D., Neaton, J. B., Yang, P., Tretiak, S. & Mathies, R. A. Excited-state Vibrational Dynamics toward the Polaron in Methylammonium Lead Iodide Perovskite. *Nat. Commun.* **9**, 2525 (2018).
- R6. Levchenko, S. V. & Krylov, A. I. Electronic Structure of Halogen-Substituted Methyl Radicals: Equilibrium Geometries and Vibrational Spectra of CH_2Cl and CH_2F . *J. Phys. Chem. A* **106**, 5169–5176 (2002).
- R7. Monserrat, B., Drummond, N. D. & Needs, R. J. Anharmonic Vibrational Properties in Periodic Systems: Energy, Electron-phonon Coupling, and Stress. *Phys. Rev. B* **87**, 144302 (2013).
- R8. Ribeiro, G. A. S., Paulatto, L., Bianco, R., Errea, I., Mauri, F. & Calandra, M. Strong Anharmonicity in the Phonon Spectra of PbTe and SnTe from First Principles. *Phys. Rev. B* **97**, 014306 (2018).
- R9. Zijlstra, E. S., Tatarinova, L. L. & Garcia, M. E. Laser-induced Phonon-phonon Interactions in Bismuth. *Phys. Rev. B* **74** (22), 220301 (2006).
- R10. Seidler, P., Kongsted, J. & Christiansen, O. Calculation of Vibrational Infrared Intensities and Raman Activities Using Explicit Anharmonic Wave Functions. *J. Phys. Chem. A* **111**, 44, 11205–11213 (2007).

- R11. Hu, S., Gao, H., Qi, Y., Tao, Y., Li, Y., Reimers, J. R., Bokdam, M., Franchini, C., Sante, D. D., Stroppa, A. & Ren, W. Dipole Order in Halide Perovskites: Polarization and Rashba Band Splittings. *J. Phys. Chem. C* **121**, 23045–23054 (2017).
- R12. Mohanty, A., Swain, D., Govinda, S., Row, T. N. G., Sarma, D. D. Phase Diagram and Dielectric Properties of $\text{MA}_{1-x}\text{FA}_x\text{PbI}_3$. *ACS Energy Lett.* **4**, 2045–2051 (2019).
- R13. Brauer, J. C., Tsokkou, D., Sanchez, S., Droseros, N., Roose, B., Mosconi, E., Hua, X., Stolterfoht, M., Neher, D., Steiner, U. et. al. Comparing the excited-state properties of a mixed-cation–mixed-halide perovskite to methylammonium lead iodide. *J. Chem. Phys.* **152**, 104703 (2020).

Reviewer #2 (Remarks to the Author):

Referee report for the manuscript "Coherent Vibrational Dynamics Reveals Lattice Anharmonicity in Organic-inorganic Halide Perovskite Nanocrystals" by T. Debnath et al., submitted to Nature Communications. Comments to the authors.

The topic of the manuscript by Debnath et al. is lead halide perovskite studied by time domain vibrational spectroscopy, an intriguing and timely subject of research.

The main idea of the manuscript is to coherently excite vibrational wave-packets by a resonant 400 nm femtosecond pump pulse, and then measure the phonon properties by recording in the time-domain the transmission of a temporally delayed probe pulse. By scanning the delay ΔT between the pump and the probe pulses and by Fourier transforming over ΔT , the authors can reconstruct the Raman spectrum of the sample under investigation.

The technique is presented and then applied to compare the phonon properties of FAPbI₃, FAPbBr₃ and MAPbBr₃. In particular, in FAPbI₃ the authors report several high intensity modes at frequencies higher than 75 cm⁻¹, which are assigned to higher harmonics of the inorganic cage fundamental modes, pointing to a strong anharmonicity in the Pb-I framework. The experimental observations are supported by all-electron DFT calculations, used to calculate the energy barrier associated with the rotation of FA moiety in bromide and iodide hybrid perovskites, which suggests a stronger interaction between the FA and PbBr sublattice with respect to PbI.

These observations are used to suggest an improved stability of the Br-based hybrid perovskites over the I-based ones.

While I find this study to be interesting, there are several key points that remain unclear or confusing and which the authors should address in a revised version of the manuscript. In addition, I believe that a stronger and more quantitative connection between experimental results and the sample anharmonic properties would be appropriate for a publication in Nature Communications. For these reasons I suggest a major revision.

We thank the reviewer for the positive feedback and the endorsement of the manuscript. We appreciate the constructive comments. We have addressed them in detail and revised the manuscript accordingly. As suggested by the referee, we have tried to provide a quantitative relation between experimental results and the anharmonic properties of PNCs with different halide compositions.

My opinion is based on the following points.

- 1) By looking at Fig. 2, it seems that there is a noisy periodic modulation of the time-dependent transient differential absorption also for wavelengths redshifted with respect to bleach maximum (figure attached). In such a sample transparent region, I would not expect to see oscillations in the time-domain (or at least, they should be less intense by several orders of magnitude). The presence of such a noise can generate artefact in the frequency domain

Raman spectrum, compromising the interpretation of all the manuscript. The authors should comment on that and report the Raman spectra extracted from the Fourier transformation over all the monitored probe wavelengths.

We thank the reviewer for raising this important issue. The noisy modulations, however, have a minimal contribution in the probe region between 500-520 nm (can be clearly seen for the time trace probed at 510 nm) due to strong contribution from the coherent phonon oscillations. Our data interpretation is based on the frequency spectra obtained from the time-dependent oscillations in the bleach region only. Therefore, we think that the noisy modulations do not have much contribution in the frequency spectra.

To address the reviewer's concern, we processed our data to remove the noisy modulations (Figure R4a) and reconstructed the Raman spectra at various probe wavelengths from blue to red regions (Figure R4b). The intensity of the Raman spectra obtained from the probe wavelengths away from the bleach are less intense by at least two orders of magnitude (due to low intense time-domain oscillations). This observation supports our interpretation of the extracted frequency spectra. We have now included the new figure in the revised manuscript to clarify the concern on the noisy modulations (page 5, 8 and SI page 1).

Figure R4. a) Transient differential absorption spectra of the FAPbBr₃ PNCs in a contour diagram. b) Corresponding FFT power spectrum at different probe wavelengths (blue to red region), computed over the first 10 ps time delay. All the spectra are plotted in the same intensity scale (for comparison) and each spectrum is vertically shifted for clarity.

2) It looks that the authors have not been able to estimate and quantify from the experimental data the value of the anharmonicities in the different samples. I believe that the lack of such an estimate greatly diminishes the value of the manuscript. Furthermore, it is not clear how much the experimentally detected effect is quantitatively linked to improved stability of the Br-based hybrid perovskites.

We thank the reviewer for raising a very constructive point. Although it is difficult to quantify the anharmonicity from the experimental data, we estimated the anharmonicity constant of different samples along the strongest Pb-X bending coordinate as described below.

The anharmonicity constant χ_e can be estimated using Morse potential (ref. R14):

$$E_n = \omega_e \left(n + \frac{1}{2} \right) - \chi_e \omega_e \left(n + \frac{1}{2} \right)^2 \dots (1)$$

Here, E_n is energy of the n^{th} state. The fundamental vibrational frequency corresponds to a transition between $n = 0$ and $n = 1$, and the corresponding energy difference can be written as,

$$E_1 - E_0 = \omega_e - 2 \omega_e \chi_e \dots (2)$$

Similarly, for the first overtone, $E_2 - E_0 = 2\omega_e - 6 \omega_e \chi_e \dots (3)$

As these two transitions along the strongest Pb-Br bending occur at 14.7 cm⁻¹ and 29.1 cm⁻¹ for FAPbBr₃, solving equation (2) and (3) yields $\chi_{e,Pb-Br} \sim 0.01$.

Similarly, the anharmonicity constant for FAPbBr_xI_{3-x} and FAPbI₃ is estimated to be $\chi_{e,Br-Pb-I} \sim 0.023$ and $\chi_{e,Pb-I} \sim 0.04$, respectively. Therefore, the experimental result suggests that the anharmonicity in FAPbBr₃ PNCs is weakened by a factor of 4 due to the interaction with the FA molecule, as compared to FAPbI₃ PNCs.

We considered only the strongest Pb-X bending coordinate for the anharmonicity calculation. This is because the ground-to-excited state displacement of the potential minima is maximized along the Pb-X bending, which is 14.7 cm⁻¹ and 10.9 cm⁻¹ for FAPbBr₃ and FAPbI₃, respectively. Therefore, these are predominately responsible for the excited state deformation. Please note that, all the frequency values are approximated to near integer values in the manuscript. The relative ground-to-excited state potential minima displacement can be determined for each Raman peak using the relation, $\sigma_R \propto \Delta^2 \omega^2$ (see Ref. R5). Here σ_R , Δ and ω are the Raman scattering cross-section, the ground-to-excited potential minima displacement and the vibrational frequency, respectively. Here, the relative displacement ratio is found to be ± 10.4 , ± 1.7 and ± 1.0 for A, B and C modes respectively, for FAPbBr₃.

While establishing a quantitative link between the experimental anharmonicity and improved stability of bromide perovskites could be an interesting subject of a future study, it is out of the scope of the present manuscript.

We have now included the above discussion in the revised manuscript (see page 12, 13 and Supplementary Note 3). We believe the estimation of anharmonicity constant in different halides certainly improves the quality of our manuscript.

3) Page 6: the authors claim that “To our surprise, the negative differential absorption signal (bleaching) shows a periodic oscillation in the time domain”. It is not a surprise detecting oscillation in the time-domain in these samples, they have been already reported in several papers: Refs. 18, 30 for example, but also “Free Carrier Emergence and Onset of Electron–Phonon Coupling in Methylammonium Lead Halide Perovskite Films” J. Am. Chem. Soc. 2017, 139, 50, 18262-18270, which is actually the first report of Raman oscillations in the time domain measured in lead halide perovskite films. All these references should be appropriately cited at this point of the manuscript.

We thank the reviewer for providing us the important reference (R4 here), which we have now cited in the revised manuscript (page 3, ref. 30). However, such oscillations were not seen in colloidal PNC samples (i.e. in solutions) prior to our observation. Indeed, we were very excited to see such observation in our NC samples.

4) In the manuscript, the authors refer to the pump as an off-resonant pulse: I believe that this is misleading, being the frequency of the pump pulse above the bandgap, it is strongly absorbed and can generate photo-carriers.

Actually, in the manuscript we refer to the pump as non-resonant pulse (not off-resonant pulse). We agree with the reviewer that it is misleading. Therefore, we now refer to the pump as an above-bandgap non-resonant pulse in the revised manuscript to avoid any confusion.

5) There is no discussion here of whether photodamage is an issue (and, in fact, it is very hard to tell since it occurs immediately upon photoirradiation). The experiment has been performed with a high frequency 400 nm pump pulse, i.e. in a spectral range where the sample absorbance is much higher with respect to the bandgap. The presence of photodamage effect can compromise the interpretation and should be accurately discussed.

All the experiments were conducted on colloidal NC solutions (not thin film) placed in a cuvette under continuous stirring. This can rule out the possibility of photodamage, and this was already mentioned in the experimental section of the manuscript: “*The samples were kept in a 2-mm cuvette and the colloidal solution was continuously stirring using a magnetic stirrer to avoid sample bleaching during the experiment.*” In addition, we have checked the absorption spectra of the colloidal solution before and after the measurements and no spectral changes were noticed.

6) The description of the experimental setup lacks several essential pieces of information: the spectrum, the time duration, the fluence, the stability of the 400 nm pump pulse as well as of the probe pulse. Furthermore, the presence of a chirped probe can modify the relative weight of different phonon modes: such effects, with appropriate references to previously published works (doi.org/10.1039/B920356G: Phys. Chem. Chem. Phys., 2010, 12, 2149-2163, 10.1021/acs.jpcclett.7b00559: J. Phys. Chem. Lett. 2017, 8, 8, 1920-1924, 10.1021/acs.jpcclett.6b03027: J. Phys. Chem. Lett. 2017, 8, 5, 966-974 and

10.1021/acs.jpcllett.9b03061: J. Phys. Chem. Lett. 2019, 10, 24, 7789-7796), should be commented.

We thank the reviewer for the suggestion. We have now included the pump and probe spectrum (see Figure R5a, b), the energy stability of the 400 nm pump and the white light probe for 1 h duration (see Figure R5c), the time duration and the pump fluence in the revised manuscript and in the SI (page 5, 14, 15 and Supplementary Figure 14).

Figure R5. a) The 800 nm laser and 400 nm pump spectrum. b) The white light probe spectrum. c) The energy stability of the 400 nm pump and white light probe pulse (at 600 nm), measured for 1 h duration.

We agree with the reviewer that the presence of the chirped probe can modify the relative weight of different phonon modes. Therefore, we analysed our data after chirp-correction as shown in the manuscript. In the revised manuscript, we have now included the discussion about the effect of the chirped probe on the phonon modes by citing the appropriate references (R15 to R18) (page 8, ref. 39-42).

7) The temporal sampling interval and the temporal sampling window are missing and should be reported in the manuscript. They are particularly critical since they determine the spectral resolution and frequency window extracted upon the conversion to the frequency domain by the Fourier transform.

We have included the temporal sampling interval (which is 50 fs) in the revised manuscript (page 15). In addition, overall temporal sampling window has already reported in the previous version (10 ps, see the corresponding FFT figure legends).

8) Equation on page 6 should read as $e^{i(n\Delta\omega)T}$, with $n=1, 2, \dots$ (terms $e^{i((n+1)\Delta\omega)T}$, $e^{i((n+2)\Delta\omega)T}$, ... are not necessary and redundant). Immediately after such Eq. “ $\Delta\omega = h/T$ ” should read as “ $\Delta\omega = 2\pi/T$ ”.

We thank the reviewer for the corrections which are now included in the revised manuscript.

References (Reviewer 2):

R14. Morse, P. M. Diatomic Molecules According to the Wave Mechanics. II. Vibrational Levels. *Phys. Rev.* **34**, 57-64 (1929).

R15. Wand, A., Kallush, S., Shoshanim, O., Bismuth, O., Kosloff, R. & Ruhman, S. Chirp effects on impulsive vibrational spectroscopy: a multimode perspective. *Phys. Chem. Chem. Phys.* **12**, 2149-2163 (2010).

R16. Gdor, I., Ghosh, T., Lioubashevski, O. & Ruhman, S. Nonresonant Raman Effects on Femtosecond Pump–Probe with Chirped White Light: Challenges and Opportunities. *J. Phys. Chem. Lett.* **8**, 1920–1924 (2017).

R17. Monacelli, L., Batignani, G., Fumero, G., Ferrante, C., Mukame, S. & Scopigno, T. Manipulating Impulsive Stimulated Raman Spectroscopy with a Chirped Probe Pulse. *J. Phys. Chem. Lett.* **8**, 966-974 (2017).

R18. Batignani, G., Ferrante, C., Fumero, G. & Scopigno, T. Broadband Impulsive Stimulated Raman Scattering Based on a Chirped Detection. *J. Phys. Chem. Lett.* **10**, 7789–7796 (2019).

Reviewer #3 (Remarks to the Author):

In the manuscript "Coherent Vibrational Dynamics Reveals Lattice Anharmonicity in Organic inorganic Halide Perovskite Nanocrystals", Debnath and co-authors have used a synergy between experiments and ab-initio quantum mechanical simulations to investigate and elucidate the origin of anharmonicity in hybrid halides perovskites, in particular FAPbI₃ and FAPbBr₃.

Investigating the peculiarities and differences in this family of materials is interesting to a broad community as it can unlock new key functionalities and applications. The manuscript is clear and well written. The experiments are well described and reproducible. The calculation method is described in details.

I recommend the paper for publication after these comments below are addressed:

Firstly, we thank the reviewer for his/her appreciations and recommendation for publication of our work.

1) The authors have not included spin-orbit coupling interactions in their calculations. Would the SOC change their results?

We thank the reviewer for bringing up this important issue related to organic inorganic-hybrid systems. Although it is documented in the literature that there's a "fortuitous" cancellation of errors arising due to neglecting spin-orbit coupling and self-interaction errors in these hybrid perovskites with GGA (R19, R20), to make sure that our conclusions are robust and not an artefact of used methodologies, we have calculated the total energies as a function of unit cell volume with and without SOC for FAPbBr₃ (unit cell) in the revised SI Figure 7 (reproduced as Figure R6 below). As can be seen, the SOC doesn't modify the structural hierarchies and hence does not alter our conclusions related to rotational energy barriers on the vibrational frequencies. We have included the discussion in the revised manuscript (page 10) and in the SI (page 4).

Figure R6. The energy variation as a function of unit cell volume in FAPbBr₃ a) without SOC and b) including SOC interactions.

2) What is the symmetry of the investigated structures? Are all with the same symmetry?

We have taken the initial symmetry from experimental XRD results of FAPbBr₃, which is pseudo-cubic with space group *Pm3m*. Following that, the initial symmetry of the substitutional structures have been kept the same as FAPbBr₃, while allowing a total relaxation of atomic positions and lattice in all calculations. This point is now added in the revised manuscript (Methods, page 15).

3) The authors have considered a specific rotation of the FA molecule. It might be that in experiments multiple rotations are present at the same time. Can molecular dynamics simulations help in clarify the role of rotation with respect to the changes in the lattice and thus of the interaction between the molecule and the inorganic framework?

We thank the reviewer for raising this issue. The considered rotation is not an arbitrary one, but is a minimum-energy path. While the free-energy barriers at finite temperature will be higher, we do not expect a qualitative change in the *difference* between FAPbI₃ and FAPbBr₃. To further elucidate the effects of halide substitution on rotational barrier faced by the FA molecules in FAPbBr_xI_{3-x}, we have constrained the FAPbI₃ lattice parameters to mimic that of FAPbBr₃ as suggested by the reviewer in Query 4.

Figure R7. Calculated minimum-energy path for the rotation of FA moiety between two stable symmetrically equivalent configurations. The results are shown for FAPbI₃ (purple square), constrained FAPbI₃ (blue triangle) and FAPbBr₃ (maroon circle).

The rotational barrier of FA moiety in this constrained lattice (~0.252 eV) is similar to FAPbBr₃ (~0.25 eV) and higher than in FAPbI₃ (0.08 eV). Further, we have calculated the rotational barrier of MA molecule in MAPbBr₃. Its comparison with the rotational barrier of FA in FAPbBr₃ is presented in Figure R8.

Figure R8. Calculated minimum-energy path for the rotation of MA and FA moiety between two stable symmetrically equivalent configurations in MAPbBr₃ (red triangle) and FAPbBr₃ (maroon circle).

We note that while the rotational energy barrier is strongly dependent on the lattice expansion caused by iodine substitution from Figure R7, the nature of molecular cation also plays a crucial role as evident from Figure R8. We have now included the discussion in the revised manuscript (page 10, 12) as well as in SI (page 5, 6).

4) The combined effects of the interaction between the inorganic framework and role of anions could be decoupled by applying strain to FAPbI₃/FAPbBr₃ and changing the volume. Can the authors comment on this and show some preliminary data?

As per the reviewer's suggestion, we have constrained the volume of FAPbI₃ to that of FAPbBr₃ to decouple the effect of volume change and FA-PbX₆ cage interactions. We note that the FA moiety faces almost the same barrier as it faces in FAPbBr₃ in this constrained

system (Figure R7), highlighting the important contribution of lattice expansion in reducing the rotational barrier of FA in the iodides. This discussion is now added in the revised manuscript (page 10).

5) In the methods, the authors write "To find out the most energetically favored geometrical configuration for $\text{FAPbBr}_x\text{I}_{3-x}$ ($x = 1.5$), we first have identified 313 the most favourable site for I substitution by scanning over all the halide (X) sites in a $2 \times 2 \times 2$ supercell...". It is not clear why they use $\text{FAPbBr}_x\text{I}_{3-x}$ as this structure is not mentioned in the manuscript and SI

The rotational barrier of FA in $\text{FAPbBr}_{1.5}\text{I}_{1.5}$ is presented in Figure 4a of the main manuscript to show the trend of change in barrier with halide substitution. The change in supercell volume as a function of Br content is also presented in Figure 4b.

6) It would be useful if the authors collect all calculations in a database and make it available in online repositories.

We thank the reviewer for reminding this important point. All the data files are uploaded to NOMAD data repository (R21) and will be made public upon the publication of the manuscript.

References:

R19. Du, M.H. Density Functional Calculations of Native Defects in $\text{CH}_3\text{NH}_3\text{PbI}_3$: Effects of Spin–Orbit Coupling and Self-Interaction Error. *J. Phys. Chem. Lett.* **6**, **8**, 1461 (2015).

R20. Umari, P., Mosconi, E., & Angelis, F. De. Relativistic GW calculations on $\text{CH}_3\text{NH}_3\text{PbI}_3$ and $\text{CH}_3\text{NH}_3\text{SnI}_3$ perovskites for solar cell applications. *Scientific Reports* **4**, 4467 (2014).

R21. <https://www.nomad-coe.eu>.

REVIEWER COMMENTS

Reviewer #1 (Remarks to the Author):

The authors have seriously and thoroughly addressed my concerns and, in my view, the points raised by the other reviewers. The new treatment of the vibrational data is more robust, and the additional discussion provides ample opportunities for further investigation. I am now happy to recommend this work for publication. My only minor comment is the new laser stability data in Supplementary Figure 14. The caption says the energy stability was measured for 1 hour, but the data suggests only 60 seconds.

Reviewer #2 (Remarks to the Author):

The authors have partially clarified some of my comments, but I still have several concerns regarding their work, with the data analysis that should be made more robust and the conclusion that are not enough strongly supported in the discussion. For these reasons I believe that the present version of the manuscript is not appropriate for Nat. Comm.

Major points:

- 1) As I previously pointed out (comment #1), one of my main concerns is related to the periodic noise modulating the time-domain data, as evident from the data recorded at probe wavelengths away from the bleach, where the transient absorption spectra should be a flat, vanishing background. In their reply the authors re-processed the data to remove the noisy modulations and they reconstructed the Raman spectra at various probe wavelengths from blue to red regions (Figure R4b). To support the validity of their spectra, they show that the intensity of the Raman spectra obtained from the probe wavelengths away from the bleach are less intense by at least two orders of magnitude. However, I believe that this argument does not exclude that the noisy modulation (whose physical/experimental origin remains unassigned) reported in the original spectra is not affecting the FFT power Raman spectra: since the background noise is in general proportional to the amplitude of the transient absorption amplitude, it is expected to decrease in intensity by several orders of magnitude for probe wavelengths away from the bleach. This check should be performed for all the samples investigated, in particular for the FAPbI₃ PNCs, whose transient absorption (Supplementary Figure 4c) show very noisy spectra.
- 2) For what concerns my comment #2, the authors have clarified in their reply that "while establishing a quantitative link between the experimental anharmonicity and improved stability of bromide perovskites could be an interesting subject of a future study, it is out of the scope of the present manuscript". However, this is at odd with what promised in the abstract, where the authors claim that their "findings [...] unveil the superior stability of Br-based PNCs over I-based PNCs", and also with the discussion in the manuscript "the dependence of this interaction on the halide nature can explain several fundamental properties of hybrid halide perovskites. In particular, it explains the lower stability of iodide (FAPbI₃) compared to bromide (FAPbBr₃) perovskites."
- 3) For what concerns the estimate of the anharmonicity constant obtained using the Morse potential, the authors have not provided any confidence interval for the extracted values, which should be evaluated taking into account for the FFT power spectra spectral resolution (this value should be reported in the manuscript, I think that taking into account for the 10 ps temporal window used for performing the FFT it should correspond to 3.33 cm⁻¹).
- 4) The anharmonicity factors have been estimated considering only the assigned A mode. This has been obtained considering a Morse potential, which is a fair approximation for extracting the single mode anharmonicity (deviation from harmonic potential as a function of the single normal coordinate), but which gives no information on the coupling between different phonons. This latter is the mechanism that may be responsible for an energy transfer between combination bands, enhancing the interaction between the FA moiety and the inorganic lattice.
- 5) The FFT power spectrum has been computed over the first 10 ps time delay. However, this does not correspond to the overall temporal sampling window, which seems longer (at least 20 ps) by looking at Figure 2c. I am quite surprised that the complete sampling window has not been exploited for computing the FFT, since it is expected to increase the spectral resolution and data quality without affecting the signal to noise ratio. The authors should comment in this respect. In

addition, I note that performing the FFT over a 10 ps temporal window results in a 3.33 cm⁻¹ spectral resolution (Ref. 38). The FFT power spectra shown in the manuscript seem to have a smoother appearance along the frequency axis. The authors should comment also in this respect.

6) For what concerns my previous comment #6 on the stability of pump and probe pulses, the authors have shown the energy of the 400 nm pump and white light probe pulse (at 600 nm), measured for 1 h duration (in this respect I have to note that the axis do not allow the reader to quantitatively evaluate stability of the pulses). I do not think that this is a good indicator of the signal to noise ratio required to quantitatively extract the FFT power spectra. The authors should report the standard deviation of 1) the pump intensity and 2) the probe spectra (all the probe wavelengths considered for extracting the FFT power spectra), with the corresponding FFT (to confirm the absence of noise, which may affect the spectral assignments).

Minor points:

1) For what concerns my previous comment #6 on the pulse chirp, the authors have clarified in the manuscript that "It is also important to note that the presence of the chirped probe can modify the relative weight of different phonon modes[39–42] and therefore, the data is analyzed after the chirp correction". As discussed in the cited literature, correcting for the probe chirp does not allow to correct for the modification of the relative weight of the different phonon modes. Hence it would be important also to report the experimentally measured value, to confirm that it barely affects the relative weights of the monitored low frequency modes.

2) Page 6, line 100: "The interaction of pump pulse with PNCs leads to the generation of wave packet in the ground state (G.S)." However, as the authors have also clarified later in the manuscript, since the pump is above the absorption edge, vibrational wave packets can be generated both on ground and excited states. This sentence should be rephrased.

3) Figure 2c shown in the present version of the manuscript has been re-processed to filter a noise periodic modulation. I believe that the original raw data should be reported in the paper (at least as SI), clearly discussing all the steps that have been adopted to clean the data.

Reviewer #3 (Remarks to the Author):

The authors have addressed all the comments from the reviewer in a very satisfactory way. I recommend the paper for publication as it is.

Manuscript NCOMMS-20-36021A, Debnath et al.

Reply to the Reviewers' comments

Reviewer #1 (Remarks to the Author):

The authors have seriously and thoroughly addressed my concerns and, in my view, the points raised by the other reviewers. The new treatment of the vibrational data is more robust, and the additional discussion provides ample opportunities for further investigation. I am now happy to recommend this work for publication. My only minor comment is the new laser stability data in Supplementary Figure 14. The caption says the energy stability was measured for 1 hour, but the data suggests only 60 seconds.

We thank the Reviewer for his/her strong support for publication of our revised manuscript.

Regarding the new laser stability data in Supplementary Figure 14 (now Supplementary Figure 19), indeed it was measured for 1 hour duration. However, by mistake we show the data for the first 60 sec only and we thank the reviewer for pointing this out. This is now corrected in the revised manuscript (SI Page 10).

Reviewer #3 (Remarks to the Author):

The authors have addressed all the comments from the reviewer in a very satisfactory way. I recommend the paper for publication as it is.

We thank the Reviewer for recommending the revised manuscript for publication.

Reviewer #2 (Remarks to the Author):

The authors have partially clarified some of my comments, but I still have several concerns regarding their work, with the data analysis that should be made more robust and the conclusion that are not enough strongly supported in the discussion. For these reasons I believe that the present version of the manuscript is not appropriate for Nat. Comm.

We again thank the Reviewer for the interesting suggestions and constructive comments, which helped us to revise and improve the manuscript.

Major points:

- 1) As I previously pointed out (comment #1), one of my main concerns is related to the periodic noise modulating the time-domain data, as evident from the data recorded at probe wavelengths away from

the bleach, where the transient absorption spectra should be a flat, vanishing background. In their reply the authors re-processed the data to remove the noisy modulations and they reconstructed the Raman spectra at various probe wavelengths from blue to red regions (Figure R4b). To support the validity of their spectra, they show that the intensity of the Raman spectra obtained from the probe wavelengths away from the bleach are less intense by at least two orders of magnitude. However, I believe that this argument does not exclude that the noisy modulation (whose physical/experimental origin remains unassigned) reported in the original spectra is not affecting the FFT power Raman spectra: since the background noise is in general proportional to the amplitude of the transient absorption amplitude, it is expected to decrease in intensity by several orders of magnitude for probe wavelengths away from the bleach. This check should be performed for all the samples investigated, in particular for the FAPbI₃ PNCs, whose transient absorption (Supplementary Figure 4c) show very noisy spectra.

We thank the reviewer again for bringing this issue. Following the reviewer's suggestion, we reconstructed the Raman spectra of FAPbBr_{3-x}I_x and FAPbI₃ PNCs at various probe wavelengths from blue to red wavelengths of the bleach maxima (Figure R2_1). The intensity of the Raman spectra obtained from the probe wavelengths away from the bleach are less intense by at least two orders of magnitude (similar to the data of FAPbBr₃ PNCs). This observation further supports our interpretation of the extracted frequency spectra and we hope that it also clarifies reviewer's concern. We have now included the new figure in the revised supplementary information (SI page 5).

Figure R2_1. FFT power spectrum at different probe wavelengths (blue to red region compared to bleach maxima) for a) FAPbBr_{3-x}I_x and b) FAPbI₃ PNCs, computed over the first 10 ps time delay. All the spectra are plotted in the same intensity scale (for comparison) and each spectrum is vertically shifted for clarity.

However, the excited-state absorption (ESA) caused by the excited state wave packet can still contribute to the transient absorption signal at probe wavelengths away from the bleach. This has been pointed out by #reviewer 1 in his/her earlier comments (referring to ref. 38 in the main manuscript) and also thoroughly addressed in one of the recent papers published in *Nature Communications* **9**, 2525 (2018) for MAPbI₃ perovskites. As shown below in Fig. R2_2, in this work the authors observed clear periodic oscillation in the excited state absorption feature in MAPbI₃ perovskites in a region away from the bleach (~200 meV redshift) and the physical origin was ascribed to the formation of the excited state coherent phonons (mainly different Pb-X vibrational modes). It seems, that a similar low intense noisy ESA signals are present away from the bleach (>530 nm) in our experiments as well. We agree with the referee that the ESA signal may

have some overlap with the bleach signal and the artefact free bleach signal could be reproduced by employing advanced data processing. In the present manuscript, we are studying three different samples grown and characterize at similar conditions and the conclusions are drawn based on the comparison of their different behavior under these similar conditions (and supported by theoretical calculations). Therefore, we do not think advanced data processing is required in the present context of the manuscript. As mentioned in our previous reply, we strongly believe that the noisy modulations have a minimal contribution in the probe region between 500-520 nm (this can be clearly seen for the time trace probed at 510 nm, kindly revisit **Fig. 2d**). We have now included the above discussions in the revised SI (Page 13) and also mentioned in the revised main manuscript (Page 8).

Figure R2_2. a) Transient excited state absorption spectrum (contour plot) shown in the 830–940 nm region of MAPbI₃ perovskite excited at 560 nm. b) Time profile measured at 840–855 nm. The residual profiles are also shown revealing underlying oscillations. Reproduced from *Nature Communications* volume 9, Article number: 2525 (2018).

2) For what concerns my comment #2, the authors have clarified in their reply that “while establishing a quantitative link between the experimental anharmonicity and improved stability of bromide perovskites could be an interesting subject of a future study, it is out of the scope of the present manuscript”. However, this is at odd with what promised in the abstract, where the authors claim that their “findings [26] unveil the superior stability of Br-based PNCs over I-based PNCs”, and also with the discussion in the manuscript “the dependence of this interaction on the halide nature can explain several fundamental properties of hybrid halide perovskites. In particular, it explains the lower stability of iodide (FAPbI₃) compared to bromide (FAPbBr₃) perovskites”.

We apologize for the misunderstanding. We wanted to convey that our findings explain the reported superior stability of bromide-based perovskites compared to iodide-based perovskites. For instance, it is well-known from the literature that FA/CsPbBr₃ NCs (stable for more than a year) exhibit much higher stability compared to FA/CsPbI₃ NCs (degrade within a month) (e.g. *J. Mater. Chem. A* 2019, 7, 16912-16919). However, following reviewer’s suggestion, we performed theoretical calculation to estimate the formation energies of FAPbBr₃ and FAPbI₃ according to the following formula:

$$E_{Form} = E_{FAPbX_3} - E_{FAX} - E_{PbX_2}$$

As expected, the bromide is found to be more stable compared to the iodide perovskite by 0.53 eV per formula unit.

Note that the conclusion about the effects of the interaction between the FA cation and the Pb-X unit is based on our combined experimental and theoretical results, and we have tried to correlate our results with the halide-dependent stability. We have now added the calculation and comparison of formation energies in the revised manuscript (in the discussion part) to support our claim (Page 13).

- 3) For what concerns the estimate of the anharmonicity constant obtained using the Morse potential, the authors have not provided any confidence interval for the extracted values, which should be evaluated taking into account for the FFT power spectra spectral resolution (this value should be reported in the manuscript, I think that taking into account for the 10 ps temporal window used for performing the FFT it should correspond to 3.33 cm⁻¹).

Yes, the reviewer is correct. The FFT spectral resolution is 3.33 cm⁻¹ for the 10 ps temporal window in this work. As suggested, FFT spectral resolution is included in the revised manuscript now (Page 15).

As our spectral resolution doesn't allow us to quantify the exact frequency values required for estimation of the anharmonicity constant using the Morse potential, we have approximated the frequency values by modeling with Gaussian functions. Although the FAPbBr₃ PNCs can nicely be modelled by two Gaussians for A and 2A modes, for the other two PNCs a third Gaussian is required to get a nice fitting (see Figure R2_3). Of course, with 3.33 cm⁻¹ spectral resolution, it would be difficult to provide a confidence interval for the estimated frequency values required for anharmonicity constant calculation. We have included the method (by stating the approximation) used for estimation of the anharmonicity constant in the revised supplementary information (SI page 9, 13).

Figure R2_3. Gaussian fitting of A and 2A modes of the FFT power spectrum of FAPbBr₃, FAPbBr_{3-x}I_x and FAPbI₃ PNCs (top to bottom).

- 4) The anharmonicity factors have been estimated considering only the assigned A mode. This has been obtained considering a Morse potential, which is a fair approximation for extracting the single mode anharmonicity (deviation from harmonic potential as a function of the single normal coordinate), but which gives no information on the coupling between different phonons. This latter is the mechanism that may be responsible for an energy transfer between combination bands, enhancing the interaction between the FA moiety and the inorganic lattice.

We thank the reviewer for raising this interesting issue. Indeed the anharmonicity factor associated with the other modes can shed light on the coupling between different phonons. In this work, the combination bands that appear in the FFT spectra qualitatively explain the interaction between different phonons (e.g. different bending and stretching modes). However due to the limited time resolution (and therefore limited FFT spectral resolution), it would be difficult to estimate the anharmonicity constant associated with high frequency modes where the intensity of the modes is fairly low. For the A mode, however, due to its strong intensity, we can estimate the anharmonicity constant value with good approximation (as described in comment 3). We believe that this would be very interesting for future studies. We have mentioned this in the revised manuscript (Page 9).

- 5) The FFT power spectrum has been computed over the first 10 ps time delay. However, this does not correspond to the overall temporal sampling window, which seems longer (at least 20 ps) by looking at Figure 2c. I am quite surprised that the complete sampling window has not been exploited for computing the FFT, since it is expected to increase the spectral resolution and data quality without affecting the signal to noise ratio. The authors should comment in this respect. In addition, I note that performing the FFT over a 10 ps temporal window results in a 3.33 cm⁻¹ spectral resolution (Ref. 38). The FFT power spectra shown in the manuscript seem to have a smoother appearance along the frequency axis. The authors should comment also in this respect.

We agree with the reviewer that computing FFT over a longer temporal window can increase the spectral resolution, however, it may affect the signal-to-noise ratio for some specific modes. For example, kindly see the following example (Figure R2_4) where the authors performed FFT in the different spectral windows. They found that an important mode (the SO phonon) remains undetected when FFT performed over a long sampling window, however, it can be clearly seen in the data obtained with a smaller sampling window. In addition, many peaks appeared for a longer sampling window and it makes it difficult to differentiate the real vs noise peaks. Therefore, we also performed FFT for a small sampling window (~10 ps) and observed reasonably good spectra with a decent signal-to-noise ratio.

Figure R2_4. FFT power spectrum computed over a 12 ps time interval of CdSe/CdS core/shell NPLs. The inset shows the FFT power spectrum computed over the first 3 ps, where the SO phonon peak is detected clearly. Reproduced with permission from *Nano Lett.* 2017, 17, 5, 3312–3319.

All the FFT power spectra reported in the manuscript are plotted after applying *Spline* in Origin for a smoother representation (see Figure R2_5 for comparison of raw data and *Spline* data). *B-Spline*, *Spline* etc. are often used during the data processing and we have adapted the same in our manuscript for better presentation of the data. We have now clearly stated this in the revised manuscript (Page 15).

Figure R2_5. FFT power spectrum of FAPbBr₃ PNCs, before and after applying *Spline* in Origin for a smoother representation.

- 6) For what concerns my previous comment #6 on the stability of pump and probe pulses, the authors have shown the energy of the 400 nm pump and white light probe pulse (at 600 nm), measured for 1 h duration (in this respect I have to note that the axis do not allow the reader to quantitatively evaluate stability of the pulses). I do not think that this a good indicator of the signal to noise ratio required to quantitatively extract the FFT power spectra. The authors should report the standard deviation of 1) the pump intensity and 2) the probe spectra (all the probe wavelengths considered for extracting the FFT power spectra), with the corresponding FFT (to confirm the absence of noise, which may affect the spectral assignments).

Previously, by mistake we only showed the first 60 sec of the laser stability instead of 1 h (Supplementary Figure 19, pointed out by the #reviewer 1 as well), which may be one of the possible reason the reviewer is unable to evaluate the stability of the pulses quantitatively. Furthermore, we noted that the attenuation of the probe pulse was too much in the previous white light probe pulse stability (at 600 nm) measurement. We have corrected the time axis in the revised manuscript (and expanded the energy axis), as well as reported the white light probe pulse stability at less attenuation (SI page 10). We hope the reviewer appreciate the laser stability.

Following reviewer's suggestion, we have now calculated the standard deviation of the 400 nm pump stability measurement which is 0.61 nJ (having average pump power 50.3 nJ, measured for 1 h i.e., ~1.2% energy fluctuation). During the previous revision, we have measured the continuum white light probe pulse stability at 600 nm only. As suggested, we have now measured the white light probe pulse energy stability (attenuated) for 60 sec at various wavelengths considered for extracting the FFT power spectra (between 450 nm to 750 nm, in the span of total ~1 h duration, see Figure R2_6 and Supplementary Fig. 19c). Furthermore, we also calculated the standard deviation of measured energy stability of the probe spectra for various wavelengths (Supplementary Fig. 19d). The standard deviation of measured energy stability of the probe spectra for various wavelengths considered for extracting the FFT power spectra indicates we have less than 1% probe energy fluctuation, and therefore further strengthen our spectral assignments regarding the corresponding FFT values.

We have now included the new measurements and a discussion on laser stability in the revised supplementary information (SI Page 10, 14).

Figure R2_6. The energy stability of a) the 400 nm pump and b) the continuum white light probe pulse (at 600 nm), measured for 1 h duration. c) The average energy (in nJ) of the attenuated continuum white light probe pulse between 450 nm to 750 nm, measured for 60 sec each, in the span of total ~1 h duration and d) the corresponding wavelength dependent standard deviation (in pJ) of the probe pulse.

Minor points:

- 1) For what concerns my previous comment #6 on the pulse chirp, the authors have clarified in the manuscript that ‘‘It is also important to note that the presence of the chirped probe can modify the relative weight of different phonon modes[39–42] and therefore, the data is analyzed after the chirp correction’’. As discussed in the cited literature, correcting for the probe chirp does not allow to correct for the modification of the relative weight of the different phonon modes. Hence it would be important also to report the experimentally measured value, to confirm that it barely affects the relative weights of the monitored low frequency modes.

We thank the reviewer for bringing this issue back. First of all, we would like to clarify that the data analysis for spectral and time-domain was performed after the chirp correction (Fig. 2). However, raw data was used to obtain the FFT power spectra. We apologize for the unclear statement which is clarified in the revised manuscript (Page 8). As suggested, we have now performed FFT for FAPbBr₃ PNCs after the chirp correction and it closely resembles that of the raw data (Figure R2_7). Moreover, one can clearly see that

the chirp correction barely affects the relative weight of the low-frequency modes by comparing the two spectra. This discussion is now incorporated in the revised manuscript and SI (Page 8 and SI Page 2).

Figure R2_7. Comparison of FFT power spectrum of the FAPbBr₃ PNCs before (raw) and after (no chirp) chirp correction.

- 2) Page 6, line 100: “The interaction of pump pulse with PNCs leads to the generation of wave packet in the ground state (G.S.)”; However, as the authors have also clarified later in the manuscript, since the pump is above the absorption edge, vibrational wave packets can be generated both on ground and excited states. This sentence should be rephrased.

As suggested, we have rephrased the sentence as following (Page 6): *The interaction of above bandgap pump pulse with the PNCs leads to generation of wave packet in the ground state (G.S.) as well as excited state (E.S.).*

- 3) Figure 2c shown in the present version of the manuscript has been re-processed to filter a noise periodic modulation. I believe that the original raw data should be reported in the paper (at least as SI), clearly discussing all the steps that have been adopted to clean the data.

As suggested, we have included the raw data and discussion about data processing in the revised SI (SI Page 1, 14).

REVIEWERS' COMMENTS

Reviewer #2 (Remarks to the Author):

The technical concerns that I raised in my previous Referral have been seriously addressed in the authors' revision, which made an excellent job, with 2 minor points that should be fixed. For what concerns the suitability of the paper for Nat. Comm., I still believe that the strength and the impact of the conclusions are more suitable to Sci. Rep. or Communications Physics, but I would like to transfer this decision to Editor.

Minor technical points:

- 1) Figure S3 and S9: as I commented in my previous referral, the background noise is in general proportional to the amplitude of the transient absorption amplitude (hence it is expected to decrease in intensity by several orders of magnitude for probe wavelengths away from the bleach). Hence the traces at different probe wavelengths reported in Fig. S3 and S9 should be normalized at the same maximum intensity.
- 2) Figure S4: in order to clarify one of my previous minor comments, the authors have shown the FFT spectra computed with and without chirp correction. This figure is irrelevant for clarifying my comment and it can be removed from the Supporting Information: as I previously pointed out, the chirp can affect the relative amplitude of the modes and "correcting for the probe chirp does not allow to correct for the modification of the relative weight of the different phonon modes" (such modification is quantitatively reported, for example, in Eq. 4 of Ref. 40 for the non-resonant case). Since the phonon modes that the authors are considering are in the low frequency spectral region ($< 150 \text{ cm}^{-1}$), a small amount of probe chirp is not expected to strongly affect the FFT spectra: the authors should simply report the measured chirp value in the Supporting Information and confirm such hypothesis.

Reply to the Reviewers' comments

Reviewer #2 (Remarks to the Author):

The technical concerns that I raised in my previous Referral have been seriously addressed in the authors' revision, which made an excellent job, with 2 minor points that should be fixed. For what concerns the suitability of the paper for Nat. Comm., I still believe that the strength and the impact of the conclusions are more suitable to Sci. Rep. or Communications Physics, but I would like to transfer this decision to Editor.

We thank the Reviewer for his/her support for publication of our revised manuscript.

Minor technical points:

1) Figure S3 and S9: as I commented in my previous referral, the background noise is in general proportional to the amplitude of the transient absorption amplitude (hence it is expected to decrease in intensity by several orders of magnitude for probe wavelengths away from the bleach). Hence the traces at different probe wavelengths reported in Fig. S3 and S9 should be normalized at the same maximum intensity.

We have reported the traces in Fig. S3 and S9 (now S8) before and after normalization (to the same maximum intensity) in the revised supplementary information, as suggested by the reviewer.

2) Figure S4: in order to clarify one of my previous minor comments, the authors have shown the FFT spectra computed with and without chirp correction. This figure is irrelevant for clarifying my comment and it can be removed from the Supporting Information: as I previously pointed out, the chirp can affect the relative amplitude of the modes and \therefore correcting for the probe chirp does not allow to correct for the modification of the relative weight of the different phonon modes \therefore (such modification is quantitatively reported, for example, in Eq. 4 of Ref. 40 for the non-resonant case). Since the phonon modes that the authors are considering are in the low frequency spectral region ($< 150 \text{ cm}^{-1}$), a small amount of probe chirp is not expected to strongly affect the FFT spectra: the authors should simply report the measured chirp value in the Supporting Information and confirm such hypothesis.

We have now removed the Figure S4 as suggested by the reviewer and updated the manuscript accordingly.